# Decentralized Local Stochastic Extra-Gradient for Variational Inequalities

**Aleksandr Beznosikov**
Innopolis University,[*] MIPT,[†] HSE University and Yandex
anbeznosikov@gmail.com

**Pavel Dvurechensky**
WIAS[‡]
pavel.dvurechensky@wias-berlin.de

**Anastasia Koloskova**
EPFL
anastasia.koloskova@epfl.ch

**Valentin Samokhin**
IITP RAS[§]
samohin.vyu@phystech.edu

**Sebastian U. Stich**
CISPA[¶]
stich@cispa.de

**Alexander Gasnikov**
MIPT, HSE University and IITP RAS
gasnikov@yandex.ru

## Abstract

We consider distributed stochastic variational inequalities (VIs) on unbounded domains with the problem data that is heterogeneous (non-IID) and distributed across many devices. We make a very general assumption on the computational network that, in particular, covers the settings of fully decentralized calculations with time-varying networks and centralized topologies commonly used in Federated Learning. Moreover, multiple local updates on the workers can be made for reducing the communication frequency between the workers. We extend the stochastic extragradient method to this very general setting and theoretically analyze its convergence rate in the strongly-monotone, monotone, and non-monotone (when a Minty solution exists) settings. The provided rates explicitly exhibit the dependence on network characteristics (e.g., mixing time), iteration counter, data heterogeneity, variance, number of devices, and other standard parameters. As a special case, our method and analysis apply to distributed stochastic saddle-point problems (SPP), e.g., to the training of Deep Generative Adversarial Networks (GANs) for which decentralized training has been reported to be extremely challenging. In experiments for the decentralized training of GANs we demonstrate the effectiveness of our proposed approach.

## 1 Introduction

In large-scale machine learning (ML) scenarios the training data is often split between many devices, such as geographically distributed datacenters or mobile devices [38]. Decentralized training methods can learn an ML model with the same accuracy as if all the data would be aggregated on one single server [54, 5]. At the same time, training in a decentralized fashion has many advantages over traditional centralized approaches in such core aspects as data ownership, privacy, fault tolerance,

---

[*]Research Center for Artificial Intelligence, Innopolis University

[†]Moscow Institute of Physics and Technology

[‡]Weierstrass Institute for Applied Analysis and Stochastics

[§]Institute for Information Transmission Problems RAS

[¶]CISPA Helmholtz Center for Information Security

36th Conference on Neural Information Processing Systems (NeurIPS 2022).

and scalability. A particular instance of the decentralized learning setting is Federated Learning (FL), where the training is orchestrated by a single device or server that communicates with all the participating client devices [62, 38]. In contrast, in fully decentralized learning (FD) scenarios the devices only communicate with their neighbors in the communication network graph with possibly arbitrary topology [54]. Thus, decentralized algorithms are important in scenarios where centralized communication is expensive, not desired, or impossible.

There have been tremendous advances recently in the development, design, and understanding of decentralized training schemes [71, 95, 86, 54, 84, 92, 90, 94, 23, 81, 50]. In particular, such aspects as data-heterogeneity [90, 78, 55], communication efficiency (through local updates [52, 44] or compression [89, 45]), and personalization [93, 8] have been studied. However, all these advances were aimed at training with single-criterion loss functions leading to minimization problems, and they do not apply to more general problem classes. For example, the training of Generative Adversarial Networks (GANs) [28] requires simultaneous competing optimization of the generator and the discriminator objectives, i.e., solving a non-convex-non-concave saddle-point problem (SPP). This problem structure makes GANs notoriously difficult to train even in the single-node setting [27, 15, 16], not talking about training over decentralized datasets [58, 69, 80].

Our goal in this paper is solving decentralized stochastic SPPs, and, more generally, decentralized stochastic Minty variational inequalities (MVIs) [64, 37]. In a decentralized stochastic MVI, the data is distributed over $M \geq 1$ devices/nodes and each device $m \in [M]$ has access to its local stochastic oracle $F_m(z, \xi_m)$ for the local operator $F_m(z) := \mathbb{E}_{\xi_m \sim \mathcal{D}_m} F_m(z, \xi_m)$. The data $\xi_m$ in the device $m$ follows an unknown distribution $\mathcal{D}_m$ that can be different for each device $m \in [M]$. The devices are connected via a communication network forming a graph such that two devices can exchange information if and only if the corresponding nodes are connected by an edge in this graph. The goal is, while respecting the communication constraints, to find cooperatively a point $z^* \in \mathbb{R}^n$ such that, for all $z \in \mathbb{R}^n$,

$$\frac{1}{M} \sum_{m=1}^M \langle \mathbb{E}_{\xi_m \sim \mathcal{D}_m} F_m(z, \xi_m), z^* - z \rangle \leq 0. \tag{1}$$

A special instance of decentralized stochastic MVIs is the decentralized stochastic SPP with local objectives $f_m(x, y) := \mathbb{E}_{\xi_m \sim \mathcal{D}_m}[f_m(x, y, \xi_m)]$:

$$\min_{x \in \mathbb{R}^{n_x}} \max_{y \in \mathbb{R}^{n_y}} \left[ f(x, y) := \frac{1}{M} \sum_{m=1}^M f_m(x, y) \right]. \tag{2}$$

The relation to VI can be seen by considering the variable $z = \begin{bmatrix} x \\ y \end{bmatrix}$ and the gradient field $F_m(z) = \begin{bmatrix} \nabla_x f_m(x, y) \\ -\nabla_y f_m(x, y) \end{bmatrix}$. In the special case when $f(x, y)$ is convex-concave, the corresponding operator $F(z) = \frac{1}{M} \sum_{m=1}^M \mathbb{E}_{\xi_m} F_m(z, \xi_m)$ is monotone. However, in the context of GANs training, where $x$ and $y$ are the parameters of the generator and the discriminator, respectively, the local losses $f_m(x, y)$ are possibly non-convex-non-concave in $x, y$ and one can not assume the monotonicity of $F$ in general, see also [21].

In this paper, we develop a novel algorithm for solving problems (1) and (2). Note that the gradient descent-ascent scheme for the problem (2) may diverge even in the simple convex-concave setting with $M = 1$ device [15]. Thus, unlike [58], we use extragradient updates [48, 37, 27] as a building block and combine them with a gossip-type communication protocol [98, 12] on arbitrary, possibly time-varying, network topologies. One of the main challenges due to the communication constraints is a "network error" induced by the impossibility of all the devices to reach the exact consensus, i.e., to have exactly the same information about the current iterate of the algorithm. Thus, each device stores a local variable, and only approximate consensus among the devices can be achieved by gossip steps [47]. Unlike other decentralized algorithms [84, 58], our method avoids multiple gossip steps per iteration, which leads to better practical performance and the possibility to work on time-varying networks. Moreover, our method allows for multiple local updates between communication rounds to reduce the communication overhead. This also makes our approach suitable for communication- and privacy-restricted FL or fully decentralized settings [101].

**Our contributions.** 1) Based on extragradient updates, we develop a novel algorithm for distributed stochastic MVIs (and, as a special case, for distributed stochastic SPPs) with heterogeneous data. Our scheme supports a very general communication protocol that covers centralized settings as in Federated Learning, fully decentralized settings, local steps in both the centralized/decentralized settings, and time-varying network topologies. In particular, we are not aware of earlier works

proposing or analyzing extragradient methods with local steps for the fully decentralized setting or decentralized algorithms for stochastic MVIs over time-varying networks.

2) Under the very general communication protocol and in the three settings of MVIs, i.e., with an operator that is strongly-monotone, monotone, or non-monotone under the Minty condition, we prove the convergence of our algorithm and give an explicit dependence of the rates on the problem parameters: characteristics of the network (e.g., mixing time), data heterogeneity, the variance of the data, number of devices, and other standard parameters. These theoretical results translate to the corresponding three settings of SPPs (strongly-convex-strongly-concave, convex-concave, non-convex-non-concave under Minty condition). All our theoretical results are valid in the important heterogeneous data regime and allow judging in a quantifiable way how different properties, e.g., data heterogeneity, the scale of the noise in the data, and network characteristics, influence the convergence rate of the algorithm. Even for decentralized settings, our results are novel for time-varying graphs and three different settings of monotonicity. See also Table 1 that gives more details on our contribution compared to the existing literature. The main challenge of our analysis is to deal with the very general assumption about the communication protocol and cope with the errors caused by the stochastic nature and heterogeneity of the data and limited information exchange between the nodes of the communication network. As a byproduct of independent interest, we analyze the stochastic extragradient method with biased oracle on unbounded domains, which was not done so far in the literature.

3) We verify our theoretical results in numerical experiments and demonstrate the practical effectiveness of the proposed scheme. In particular, we train the DCGAN [79] architecture on the CIFAR-10 [51] dataset.

## 1.1 Related Work

The research on MVIs dates back at least to 1962 [64] with the classical book [41] and the recent works [59, 56, 13, 21]. VIs arise in a broad variety of applications: image denoising [25, 14], game theory and optimal control [26], robust optimization [9], and non-smooth oprimization via smooth reformulations [74, 73]. In ML, MVIs and SPPs arise in GANs training [19, 15, 16], reinforcement learning [76, 36], and adversarial training [60].

**Extragradient.** The extragradient method (EGM) was first proposed in [48], generalized as the mirror-prox method for deterministic problems in [73] and for stochastic problems with bounded variance in [37]. Yet, if the stochastic noise is not uniformly bounded, the EGM may diverge, see [15, 66].

| Reference | base method | arbitrary network | time-varying | local updates | no multiple gossip steps | SM | M | NM |
|---|---|---|---|---|---|---|---|---|
| Liu et al. 2019 [58] | Stoch. ES | ✔ | ✗ | ✗ | ✗ | ✗ | ✗ | ✔[†] |
| Beznosikov et al. 2021[11] Alg. 2 | Stoch. ES | ✔ | ✗ | ✗ | ✗ | ✔ | ✔ | ✗ |
| Barazandeh et el. 2021 [6] | Stoch. ES | ✔ | ✗ | ✗ | ✗ | ✗ | ✗ | ✔ |
| Liu et al. 2019 [59] | Deter. prox | ✔ | ✗ | ✗ | ✔ | ✗ | ✗ | ✔ |
| Mukherjee and Chakraborty 2020 [69] | Deter. ES | ✔ | ✗ | ✗ | ✔ | ✔ | ✔ | ✗ |
| Tsaknakis et al. 2020 [91] | Stoch. DA | ✔ | ✗ | ✗ | ✔ | ✗ | ✗ | ✔[‡] |
| Rogozin et al. 2021[80] | Deter. ES | ✔ | ✗ | ✗ | ✔ | ✗ | ✔ | ✗ |
| Xian et al. 2021 [97] | Stoch. DA | ✔ | ✗ | ✗ | ✔ | ✗ | ✗ | ✔[‡] |
| Beznosikov et al. 2021 [11] Alg 3 | Stoch. ES | ✗ | ✗ | ✔ | -[§] | ✔ | ✗ | ✗ |
| Deng and Mahdavi 2021[20] | Stoch. DA | ✗ | ✗ | ✔ | - | ✔ | ✗ | ✔[‡] |
| Hou et al. 2021 [32] | Stoch. DA | ✗ | ✗ | ✔ | - | ✔ | ✗ | ✗ |
| Ours | Stoch. ES | ✔ | ✔ | ✔ | ✔ | ✔ | ✔ | ✔ |

[†] – homogeneous case, [‡] – non-convex-concave SPP (other works use minty condition – (NM)), § – this column does not apply to centralized algorithms.

Table 1: Comparison of approaches for distributed strongly-monotone (SM), monotone (M), and non-monotone (NM) VIs or, respectively, strongly-convex-strongly-concave, convex-concave, non-convex-non-concave SPPs. Definitions of columns: **base method** — the non-distributed algorithm that is taken as the basis for the distributed method, typically it is either the extragradient method (EGM) or the descent-ascent (DA); **arbitrary network** — supporting fully decentralized vs. only centralized topology; **time-varying** — decentralized method supporting time-varying network topology; **local updates** — method supporting local steps between communications; **no multiple gossip steps** — at one global iteration the method does not use many iterations of gossip averaging to reach a good consensus accuracy; **SM, M, NM** — monotonicity assumption, see Assumption 3.2.

**Decentralized algorithms for MVIs and SPPs** are the most closely related to our work. In Table 1, we summarize their features and make a comparison with our algorithm, showing that, e.g., existing methods do not support arbitrary time-varying network typologies. The methods that use multiple rounds of gossip averaging (sparse communication) per iteration [58, 11, 6] can give near-optimal theoretical rates, but are often unstable in practice. Thus, it is preferable to have only one sparse

communication per iteration [59, 69, 91, 80, 97]. The second column of the table refers to standard algorithms that are extended to distributed settings in the corresponding work. In particular, the algorithm of [59] requires expensive proximal updates. The closest work to ours is [11], where a decentralized EGM without local steps is analyzed in the (strongly-)monotone setting. Unlike our more general algorithm with local steps, theirs require multiple gossip updates in each iteration which is not desired in practice. For the FL, i.e., centralized, setting, [11] studies the EGM with local steps in the strongly-monotone setting, and [20, 32] study the descent-ascent method with local steps. Yet, all three works do not consider arbitrary time-varying graphs as in our work.

## 2 Algorithm

In this section, we present and discuss the proposed algorithm (Algorithm 1) that is based on two main ideas: (i) the extragradient step, as in the classical methods for VIs [48, 73], and (ii) the gossip averaging [12, 71] widely used in decentralized optimization methods and in the literature on diffusion strategies in distributed learning [82, 83, 99, 2, 61]. Unlike these papers that propose algorithms for optimization problems by exploiting gradient descent, our algorithm is based on the extragradient method and is designed to solve VIs and SPPs. Moreover, unlike the mentioned works, our method also allows for local steps in-between the communication rounds and for time-varying networks and has non-asymptotic theoretical convergence rate guarantees.

Each step of Algorithm 1 can be divided into two phases. The local phase (lines 4–6) consists of a step of the stochastic extragradient method at each node using only local information. As in the non-distributed case, the nodes first make an extrapolation step "to look into the future" and then an update based on the operator value at the "future" point. This is followed by the communication phase (gossip step) (line 7), during which the nodes share and average local iterates with their neighbors $\mathcal{N}_m^k$ in the communication network graph corresponding to the iteration $k$. The averaging process involves the weights $w_{m,i}^k$ which are the elements of the matrix $W^k$ called the mixing matrix:

**Definition 2.1** (Mixing matrix). We call a matrix $W \in [0; 1]^{M \times M}$ a mixing matrix if it satisfies the following conditions: 1) $W$ is symmetric, 2) $W$ is doubly stochastic ($W\mathbf{1} = \mathbf{1}$, $\mathbf{1}^T W = \mathbf{1}^T$, where $\mathbf{1}$ denotes the vector of all ones), 3) $W$ is aligned with the network: $w_{ij} \neq 0$ if and only if $i = j$ or the edge $(i, j)$ is in the communication network graph.

Reasonable choices of mixing matrices are, for example, (i) $W^k = I_M - \frac{L^k}{\lambda_{\max}(L^k)}$, where $L^k$ is the Laplacian matrix of the network graph at the step $k$ and $I_M$ is the identity matrix, or (ii) using some local rules in the graph, based on the degrees of the neighboring nodes [98]. Note that our setting has a great flexibility since in-between the iterations the topology of the communication graph is allowed to change, and the matrix $W^k$, that encodes the current structure of the network, changes accordingly. This is encoded in line 2, where the matrix $W^k$ is generated by some rule $\mathcal{W}^k$ which can have different nature. Examples include deterministic choice of a sequence of matrices $W^k$, sampling from a time-varying probability distribution on matrices. Even local steps without communication can be encoded with a diagonal matrix $W^k$.

---

**Algorithm 1** Extra Step Time-Varying Gossip Method

---

**parameters:** stepsize $\gamma > 0$, $\{\mathcal{W}^k\}_{k \geq 0}$ – rules or distributions for mixing matrix in iteration $k$.
**initialize:** $z^0 \in \mathcal{Z}, \forall m : z_m^0 = z^0$
1: **for** $k = 0, 1, 2, \ldots$ **do**
2:     Sample matrix $W^k$ from $\mathcal{W}^k$
3:     **for** each node $m$ **do**
4:         Generate independently $\xi_m^k \sim \mathcal{D}_k$, $\xi_m^{k+1/3} \sim \mathcal{D}_k$
5:         $z_m^{k+1/3} = z_m^k - \gamma F_m(z_m^k, \xi_m^k)$
6:         $z_m^{k+2/3} = z_m^k - \gamma F_m(z_m^{k+1/3}, \xi_m^{k+1/3})$
7:         $z_m^{k+1} = \sum_{i \in \mathcal{N}_m^k} w_{m,i}^k z_i^{k+2/3}$
8:     **end for**
9: **end for**

---

To ensure that it is possible to approach the consensus between the nodes, we need the following assumption on the mixing properties of the matrix sequence $W^k$.

**Assumption 2.2** (Expected Consensus Rate). We assume that there exist a constant $p \in (0, 1]$ and an integer $\tau \geq 1$ such that, after $K$ iterations, for all matrices $Z \in \mathbb{R}^{d \times M}$ and all integers $l \in \{0, \ldots, K/\tau\}$,

$$\mathbb{E}_W[\|ZW_{l,\tau} - \bar{Z}\|_F^2] \leq (1-p)\|Z - \bar{Z}\|_F^2, \tag{3}$$

where $W_{l,\tau} = W^{l\tau} \cdot \ldots \cdot W^{(l+1)\tau-1}$, we use the matrix notation $Z = [z_1, \ldots, z_M]$, $\bar{Z} = [\bar{z}, \ldots, \bar{z}]$ with $\bar{z} = \frac{1}{M}\sum_{m=1}^M z_m$, and the expectation $\mathbb{E}_W$ is taken over distributions of $W^t$ and indices $t \in \{l\tau, \ldots, (l+1)\tau - 1\}$.

This assumption ensures that, after $\tau$ gossip steps with such time-varying matrices, we improve the consensus between the nodes, i.e., how close each $z_m$ is to $\bar{z}$, by the factor of $\frac{1}{1-p}$. Importantly, in this case, some matrices $W^k$ can be, for example, the identity matrix (which corresponds to performing only local steps in iteration $k$).

Assumption 2.2 has been recently quite popular in the literature on distributed optimization methods [72, 44, 49]. Moreover, it is very general and covers many special cases of decentralized and centralized algorithms. For example, if we fix $W^k = W$ for some fixed connected graph, we get a decentralized algorithm on this graph. If, at the same time, we set the matrix $W = \frac{1}{M}\mathbf{1}\mathbf{1}^T$, then it is easy to see that we get an analog of the centralized setting with the averaging over all nodes performed in each communication step. If we take $W^k = W$ for some fixed connected graph at every $\tau$-th step and in other steps use $W^k = I_M$, we have a decentralized (and, in particular, centralized) algorithm with local steps [87, 30, 44] and communications after each $\tau$ iterations. Generic Assumption 2.2 covers also many other settings of time-varying decentralized topologies, e.g., random topologies, cliques, $B$-connected graphs [35, 70]. Below we show that, under an appropriate choice of the stepsize, our extragradient method provably converges under such a general assumption that covers centralized and decentralized settings, local steps in both centralized and decentralized settings, and changing topologies of the communication graph. Even for decentralized settings, this is novel for time-varying graphs and three different settings of monotonicity which we consider.

# 3 Setting and Assumptions

In this section, we introduce necessary assumptions that are used to analyze the proposed algorithm.

**Assumption 3.1** (Lipschitzness). For all $m$, the operator $F_m(z)$ is Lipschitz with constant $L$, i.e.,

$$\|F_m(z_1) - F_m(z_2)\| \leq L\|z_1 - z_2\|, \quad \forall z_1, z_2. \tag{L}$$

This is a standard assumption that is used in the analysis of all the methods displayed in Table 1.

**Assumption 3.2.** We consider three scenarios for the operator $F$, namely, when $F$ is strongly-monotone, monotone and non-monotone, but with an additional assumption:
**(SM) Strong monotonicity.** There exists $\mu > 0$ such that, for all $z_1, z_2$,

$$\langle F(z_1) - F(z_2), z_1 - z_2 \rangle \geq \mu\|z_1 - z_2\|^2. \tag{SM}$$

**(M) Monotonicity.** For all $z_1, z_2$, it holds that:

$$\langle F(z_1) - F(z_2), z_1 - z_2 \rangle \geq 0. \tag{M}$$

**(NM) Non-monotonicity (Minty).** There exists $z^*$ such that, for all $z$,

$$\langle F(z), z - z^* \rangle \geq 0. \tag{NM}$$

Assumptions (SM), (M) and (L) are standard and classical assumptions in the literature on VIs. Assumption (NM) is sometimes called the *Minty or Variational Stability condition* and it has been widely used recently by the community as a structured variant of non-monotonicity [18, 34, 63, 59, 39, 33, 21], particularly, since it is appropriate in GANs training [57, 58, 22, 6].

The next assumption is standard for the stochastic setting.

**Assumption 3.3** (Bounded noise). $F_m(z, \xi)$ is unbiased and has bounded variance, i.e., for all $z$,

$$\mathbb{E}[F_m(z, \xi)] = F_m(z), \quad \mathbb{E}[\|F_m(z, \xi) - F_m(z)\|^2] \leq \sigma^2. \tag{4}$$

Our last assumption reflects the variability of the local operators compared to their mean and is usually called $D$-heterogeneity. This assumption is widely used in the analysis of local-steps (and not only) algorithms for minimization problems [40, 96, 30, 85, 4, 1, 31, 17, 24]. Moreover, [20, 32] use this assumption for the analysis of centralized local-steps methods for SPPs. The authors of [58] assume $D = 0$ for the decentralized training of GANs. Even in this case algorithms' analysis can be challenging.

**Assumption 3.4** ($D$-heterogeneity). *The values of the local operator have bounded variablility, i.e., for all $z$,*

$$\|F_m(z) - F(z)\|^2 \le D^2. \tag{5}$$

## 4 Main Results

In this section, we present the convergence rate results for the proposed method under different settings of Assumption 3.2. To present the main result, we introduce notation $\bar{z}^k := \frac{1}{M}\sum_{m=1}^{M} z_m^k$, $\bar{z}^{k+1/3} := \frac{1}{M}\sum_{m=1}^{M} z_m^{k+1/3}$ for the averaged among the devices iterates and $\hat{z}^k = \frac{1}{k+1}\sum_{i=0}^{k} \bar{z}^{i+1/3}$ for the averaged among the devices and iterates sequence, a.k.a. ergodic average. Finally, we denote $\Delta = \frac{\tau}{p}\left(\frac{D^2\tau}{p} + \sigma^2\right)$ which plays the role of the consensus error, i.e., the error caused by the impossibility of reaching the exact consensus between the nodes. Note that the data heterogeneity appears in the convergence rates only through the quantity $\Delta$.

**Theorem 4.1** (Main theorem). *Let Assumptions 2.2, 3.1, 3.3, 3.4 hold and the sequences $\bar{z}^k$, $\hat{z}^k$ be generated by Algorithm 1 that is run for $K > 0$ iterations. Then,*

- ***Strongly-monotone case:*** *under Assumption 3.2(SM), with $\gamma = \tilde{\mathcal{O}}\left(\min\left\{\frac{p}{\tau L}, \frac{1}{\mu K}\right\}\right)$ it holds that*

$$\mathbb{E}\left[\|\bar{z}^{K+1} - z^*\|^2\right] = \tilde{\mathcal{O}}\left(\|z^0 - z^*\|^2 \cdot \exp\left(-\frac{\mu K p}{240 L \tau}\right) + \frac{\sigma^2}{\mu^2 M K} + \frac{L^2\Delta}{\mu^4 K^2}\right); \tag{6}$$

- ***Monotone case:*** *under Assumption 3.2(M), for any convex compact $\mathcal{C}$ s.t. $z^0, z^* \in \mathcal{C}$ and $\max_{z,z'\in\mathcal{C}}\|z - z'\| \le \Omega_{\mathcal{C}}$, with $\gamma = \mathcal{O}\left(\min\left\{\frac{1}{L}, \left(\frac{\Omega_{\mathcal{C}}^2 M}{K\sigma^2}\right)^{\frac{1}{2}}, \left(\frac{\Omega_{\mathcal{C}}^2}{K^2 L^2\Delta}\right)^{\frac{1}{4}}\right\}\right)$ it holds that*

$$\sup_{z\in\mathcal{C}}\mathbb{E}\left[\langle F(z), \hat{z}^K - z\rangle\right] = \mathcal{O}\left(\frac{L\Omega_{\mathcal{C}}^2}{K} + \frac{\sigma\Omega_{\mathcal{C}}}{\sqrt{MK}} + \frac{\sqrt{L\Omega_{\mathcal{C}}^3\sqrt{\Delta}}}{\sqrt{K}} + \sqrt{\frac{(\Delta + L^2\Omega_{\mathcal{C}}^2)\Omega_{\mathcal{C}}\sqrt{\Delta}}{KL}}\right). \tag{7}$$

*Under the additional assumption that, for all $k$, $\|\bar{z}^k\| \le \Omega$, with $\gamma = \mathcal{O}\left(\min\left\{\frac{1}{L}, \left(\frac{\Omega_{\mathcal{C}}^2 M}{K\sigma^2}\right)^{\frac{1}{2}}, \left(\frac{\Omega_{\mathcal{C}}^2}{K^2 L^2\Delta}\right)^{\frac{1}{4}}, \left(\frac{\Omega_{\mathcal{C}}^2}{K((\Omega+\Omega_{\mathcal{C}})L\sqrt{\Delta}+\Delta)}\right)^{\frac{1}{2}}\right\}\right)$, we have that*

$$\sup_{z\in\mathcal{C}}\mathbb{E}\left[\langle F(z), \hat{z}^K - z\rangle\right] = \mathcal{O}\left(\frac{L\Omega_{\mathcal{C}}^2}{K} + \frac{\sigma\Omega_{\mathcal{C}}}{\sqrt{MK}} + \frac{\sqrt{L\Omega_{\mathcal{C}}^3\sqrt{\Delta}}}{K^{3/4}} + \sqrt{\frac{((\Omega+\Omega_{\mathcal{C}})L\sqrt{\Delta}+\Delta)\Omega_{\mathcal{C}}^2}{K}}\right); \tag{8}$$

- ***Non-monotone case:*** *under Assumption 3.2(NM) and if $\|z^0\| \le \Omega, \|z^*\| \le \Omega$, with $\gamma = \mathcal{O}\left(\min\left\{\frac{1}{L}, \left(\frac{\Omega^2}{K^2 L^2\Delta}\right)^{\frac{1}{4}}\right\}\right)$:*

$$\mathbb{E}\left[\frac{1}{K+1}\sum_{k=0}^{K}\|F(\bar{z}^k)\|^2\right] = \mathcal{O}\left(\frac{L^2\Omega^2}{K} + \frac{\sigma^2}{M} + L\Omega\sqrt{\Delta} + \frac{\sqrt{L\Omega\Delta^{3/4}}}{\sqrt{K}}\right). \tag{9}$$

*Under the additional assumption that, for all $k$, $\|\bar{z}^k\| \le \Omega$, with $\gamma = \mathcal{O}\left(\min\left\{\frac{1}{L}, \left(\frac{\Omega^2}{KL\Delta}\right)^{\frac{1}{3}}\right\}\right)$, we have that*

$$\mathbb{E}\left[\frac{1}{K+1}\sum_{k=0}^{K}\|F(\bar{z}^k)\|^2\right] = \mathcal{O}\left(\frac{L^2\Omega^2}{K} + \frac{\sigma^2}{M} + \frac{(L\Omega\Delta)^{2/3}}{K^{1/3}} + L\Omega\sqrt{\Delta}\right). \tag{10}$$

The proof of the theorem is given in the supplementary material, where one can also find explicit dependence of the rates on the stepsize $\gamma$ before it is chosen optimally. We underline that the standard analysis [37] does not apply for the following reasons. Firstly, unlike [37], in our problem (1), the feasible set is not bounded, which is especially important for the analysis in the monotone and

non-monotone settings. Secondly, our algorithm has an additional communication step (line 7) between the computational nodes, which leads to the impossibility for all the nodes to have the same information about the global operator $F(z)$ and about the current iterate $z$. This, in order, leads to a biased oracle that, unlike existing works, has to be analyzed in the setting of an unbounded feasible set, which is quite challenging. To analyze our variant of the extragradient method, we successfully handle this challenge. Our key steps are to bound the bias (see, e.g., the last two terms in the r.h.s. of Lemma C.8 that are caused by the network errors), prove the boundedness in expectation of the sequence of the iterates for monotone (see Section C.3.1 of the supplementary material) and non-monotone (see Section C.4.1 of the supplementary material) cases, which may be of independent interest and which we have not seen in the literature, even in the non-distributed setting with biased stochastic oracles. Proving the boundedness is challenging due to the noise caused by the stochasticity and heterogeneity of the data and network effects due to the imperfect exchange of information. Surprisingly, in the end, we still manage to analyze our algorithm under the very general Assumption 2.2 and we are not aware of any results with similar generality of the settings: different network topologies (including time-varying), distributed architectures, different monotonicity assumptions.

The provided convergence rates have an explicit dependence on the problem parameters: the network that is characterized by the mixing time $\tau$ and the mixing factor $p$, the data heterogeneity $D$ (these three quantities appear in the convergence rates only through the quantity $\Delta$), the variance $\sigma^2$ of the noise in the data, the Lipschitz constant $L$, the strong monotonicity parameter $\mu$, the number of nodes/devices $M$. Thus, our rates allow judging how different properties, e.g., data heterogeneity, noise level, and network characteristics influence the convergence rates. This, in particular, opens up an opportunity for a meta-optimization process if we can design the network and change $M, \tau, p$ to achieve faster convergence.

We now discuss the convergence results obtained in the theorem, and also compare them with already existing algorithms (see Table 1) and their guarantees. Firstly, all the estimates have a similar several-term structure. The first term corresponds to the deterministic setting and is similar to existing methods for smooth VIs in the non-distributed setting. Only in the strongly-convex case, there is an additional factor $\tau/p$ that increases the condition number $L/\mu$ of the problem. The second (stochastic) term is also standard for the non-distributed setting and corresponds to the stochastic nature of the problem. Note that, for a very general distributed setting, we have managed to obtain the corresponding terms similar to the non-distributed setting. Moreover, we can see the benefit of exploiting distributed computations: the leading stochastic term depends on $\sigma^2/M$ that decreases as the number $M$ of the nodes increases. The other terms correspond to the consensus error $\Delta$ and are due to the imperfect communications between the nodes, i.e., that all the nodes can't have exactly the same information about the current iterate. Importantly, in all the cases, this error does not make the overall convergence worse since the dependence on $K$ is no worse in these terms than the dependence on $K$ in the stochastic term. In the experimental section, we illustrate that the network error is not an artifact of the analysis but is indeed present in practice.

Theorem 4.1 is formulated for a fixed budget of iterations $K$ and the corresponding stepsizes $\gamma$ that depend on $K$, which is pretty standard in the literature [37, 88, 10], where many algorithms fix the stepsize depending on the budget of the iterations. In Section D of the supplementary material, we present a simple restarting procedure that allows to extend the results of Theorem 4.1 to any-time convergence without a-priori fixing $K$. The idea is to set $K_t = 2^t$ for $t = 0, 1, \ldots$ and restart the algorithm after each $K_t$ iterations. We next make refined comments for each particular setting of monotonicity.

• **Strongly-monotone case:** In the centralized setting with local updates, our rate is slightly better than in [11]. Unlike our algorithm, centralized algorithms with local steps for SPPs in [20, 32] are based on the gradient descent-ascent method that may diverge in the stochastic setting even for bilinear problems. Moreover, their analysis implies a very small stepsize $\gamma \sim \frac{\mu p}{L^2 \tau}$ (cf. ours $\gamma \sim \frac{p}{L\tau}$), which greatly slows down the convergence of the algorithm.

For the decentralized setting, [11] propose an optimal algorithm with the rate matching the lower bound which they also give. Our rate is worse probably because of the generality of the Assumption 2.2. On the other hand, our algorithm is more practical since it avoids using multiple gossip steps at each iteration. Also, our algorithm is more general, allowing us to work with time-varying topologies and local steps even in the decentralized setting.

• **Monotone case:** The quantity $\sup_{z\in\mathcal{C}}\mathbb{E}\left[\langle F(z),\widehat{z}^K-z\rangle\right]$ in the convergence rates estimates reflects the stochastic nature of the problem and is a counterpart of the standard restricted gap (or merit) function [75]: $\mathrm{Gap}_\mathcal{C}(u):=\sup_{z\in\mathcal{C}}\left[\langle F(z),u-z\rangle\right]$. When $F$ is a monotone operator, if $\mathrm{Gap}_\mathcal{C}(\hat{u})=0$ and $\mathcal{C}$ contains a neighborhood of $\hat{u}$, then [75, 3] $\hat{u}$ is a solution to (1) and even more: it is a strong solution to the corresponding variational inequality, i.e., for all $z$, $\langle F(\hat{u}),\hat{u}-z\rangle\leq 0$. Thus, $\mathrm{Gap}_\mathcal{C}(u)$ is an appropriate measure of suboptimality in this setting and (7) guarantees that after a sufficient number of iterations, we obtain an approximate solution in expectation. Importantly, for (7), neither $z$ nor $\bar{z}^k$ are assumed to be bounded. As in the previous works on non-distributed algorithms for MVIs [75, 3], we use $\mathrm{Gap}_\mathcal{C}(u)$ with an arbitrary compact set $\mathcal{C}$ that contains $z^0$ and $z^*$ (this can be a large set). Further, (8) is a refined version of the general result (7) under the additional assumption of the boundedness of the averaged iterates. If the boundedness does not hold, we still have (7). Moreover, (7) and (8) hold for the same method, and to run the algorithm, there is no need to know in advance whether the generated sequence is bounded or not.

Only [11, 80] consider MVIs with monotone operator in distributed setting. Our algorithm is more general than theirs: our algorithm supports time-varying networks and local steps between communications. The algorithm in [11] uses multiple gossip steps between the updates of the iterates. On the one hand, this allows decreasing the consensus error $\Delta$. On the other hand, this leads to an additional factor in the number of communications compared to our estimates: the first term in their bound is $\sqrt{\chi}$ times larger than ours, where $\chi > 1$ is some condition number of the mixing matrix. Moreover, multiple gossip steps may be impractical if the communication is performed through unstable channels or is expensive for some reason. The paper [80] considers only deterministic setting.

• **Non-monotone case:** The same as in the previous case remark on the boundedness of $\bar{z}^k$, $z^*$ assumed to obtain (10) applies in this case. Further, in this setting, the convergence is guaranteed up to some accuracy that is governed by the stochastic nature of the problem (the $\sigma^2$-term) and by the distributed nature of the problem (the $\Delta$-terms). With this respect, the results are similar to the non-distributed stochastic extragradient method [7] and the distributed method [58] analyzed in the homogeneous case $D = 0$. To the best of our knowledge, convergence up to arbitrarily small accuracy can be guaranteed only for deterministic distributed methods [59], i.e., in a much simpler setting than ours. Moreover, the methods of [59] are not the most robust since they require evaluating the proximal operator of a function and it is assumed that this can be done in a closed form, which is computationally expensive and may not hold in practice.

Note that, based on our result, it is possible to achieve convergence up to arbitrarily small accuracy if one considers the homogeneous case with $D = 0$. Indeed, choosing the right batch size, for example, proportionally to $K^\alpha$ with $\alpha > 0$, one can replace $\sigma^2$ by $\frac{\sigma^2}{K^\alpha}$ in (9) and (10) and get convergence guarantees.

# 5 Experiments

In this section, we present two sets of experiments to validate the performance of Algorithm 1. In Section 5.1, we verify the obtained convergence guarantees on two examples: a strongly-monotone and a monotone bilinear problems, and in Section 5.2, we explore the non-monotone case with an application to GANs training. Extended details of the experimental setup can be found in the supplementary material.

## 5.1 Verifying Theoretical Convergence Rate

First, we focus on the verifying whether the actual behaviour of Algorithm 1 is predicted by the theoretical convergence rate in Theorem 4.1.

**Setup.** We consider a distributed bilinear SPP (2) with the objective functions $f_m(x,y) = \frac{a}{2}\|x\|^2 + bx^\top y - \frac{a}{2}\|y\|^2 + c_m^\top x$, where $x,y,c_m \in \mathbb{R}^n$, $a,b \in \mathbb{R}$ and $m \in \{1,\ldots,M\}$. This set of functions satisfy Assumptions 3.1, 3.2, 3.4 with constants $\mu = a, L^2 = a^2 + b^2$, $D = \max_m \|c_m - \bar{c}\|$. In this section, we use a ring topology on $M = 20$ nodes with uniform averaging weights, and we set the dimension $n = 5$, $b = 1$, $D \approx 3$, and keep $\tau = 1$. The value of the parameter $p$ in this setting is approximately 0.288 [46, Table 1]. To satisfy Assumption 3.3, we generate stochastic gradients by adding to the exact gradients unbiased Gaussian noise with variance $\sigma^2$.

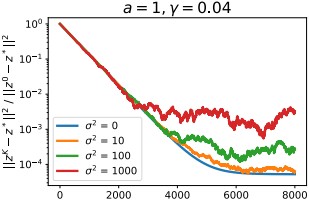 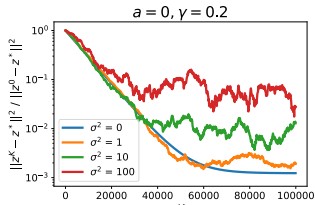

Figure 1: Convergence of Algorithm 1 with constant stepsize in the presence of stochastic noise in strongly-monotone (left) and monotone (right) cases. We observe linear convergence up to an error floor depending on the noise variance and problem parameters (cf. Theorem 4.1). In Section A.2 of the supplementary material we show convergence to arbitrary accuracy with decreasing stepsizes.

**Convergence Behaviour.** In Figure 1, we show the convergence of Algorithm 1 with a fixed stepsize on the strongly-monotone ($a = 1$) and monotone ($a = 0$) instances. In the strongly-monotone case, we see a linear convergence up to some level defined by the heterogeneity parameter and the noise. The convergence for the non-strongly-monotone problem is slower, but we also see a linear convergence up to some level (for bilinear problems this behavior is expected from the theoretical point of view [48]). Note that the convergence to some limiting accuracy is expected since when a constant stepsize is used in stochastic optimization/stochastic variational inequalities with strong convexity/monotonicity, algorithms are usually guaranteed to converge only to a vicinity of the solution, see, e.g., Theorem 2 in [66]. This is also in accordance with Theorem 4.1 that, for a fixed stepsize, guarantees the convergence to some non-zero limit accuracy and says that, to achieve the zero error, one needs to choose a decreasing stepsize. We additionally validate in Section A.2 of the supplementary material that with a decreasing stepsize, the algorithm can converge to the zero error.

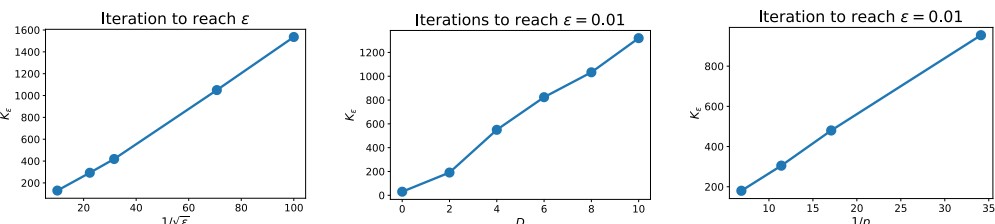

Figure 2: Verifying the $\mathcal{O}\left(\frac{D^2}{p^2 K^2}\right)$ convergence rate for the strongly-monotone noiseless ($\sigma^2 = 0$) case.

**Dependence on the Heterogeneity parameter** $D$**.** In the second set of experiments, we aim to verify the dependence on the data heterogeneity parameter $D$. Therefore, we consider the setting when $\sigma^2 = 0$. From our theory, equation (6), we predict that the most significant term in the convergence rate when $\sigma^2 = 0$ scales as $\mathcal{O}\left(\frac{D^2}{p^2 K^2}\right)$ (since the primary goal of this experiment is to study the dependence on $p$, $D$, $K$, we omit all the other fixed parameters for simplicity). We take $b = 1$, $a = 1$ and conduct experiments with the number of iterations needed to achieve the error $\frac{1}{M}\sum_{m=1}^{M}\|z_m^k - z^*\|^2 < \varepsilon$, for different $\varepsilon$. In all these experiments, the stepsize is tuned individually.

First, we verify the power of $K$ in the bounds. For this experiment, we keep $D, p$ constant and vary the accuracy $\varepsilon$. As we can see from the leftmost subplot in Figure 2, the number of iterations scales as $K \propto \frac{1}{\sqrt{\varepsilon}}$, confirming the predicted $\mathcal{O}\left(\frac{1}{K^2}\right)$ dependency of the error on $K$. Next, we measure the number of iterations sufficient to reach the error $\varepsilon = 0.01$ while varying $D$. The middle plot shows that the number of iterations scales proportionally to $D$ (showing $D \propto K$). Lastly, we depict the number of iterations to reach $\varepsilon = 0.01$ while changing the graph parameter $p$ and again observe $\frac{1}{p} \propto K$. Summarizing, these experiments verify the $\mathcal{O}\left(\frac{D^2}{p^2 K^2}\right)$ term in the convergence rate.

### 5.2 Training GANs

Our algorithm allows combining in the distributed learning setting different communication graph topologies, as well as local steps. Thus, our goal in this section is to illustrate this empirically with the experiments on GANs training. In Section A.1 of the supplementary material, we discuss to what extent our theoretical results hold for GANs training.

**Data and model.** We consider the CIFAR-10 [51] dataset containing 60000 images, equally distributed over 10 classes. We increased the size of the dataset by 4 times using transformations and adding noise. We simulate a distributed setup of 16 nodes on two GPUs and use Ray [67]. To emulate the heterogeneous setting, we partition the dataset into 16 subsets. For each subset, we select a major class that forms 20% of the data, while the rest of the data split is filled uniformly by the other classes. As a basic architecture we choose DCGAN [79], conditioned by class labels, similarly to [65] (the network architecture can be found in Section A.1). We chose Adam [42] as the optimizer. We make one local Adam step and then one gossip averaging step with time-varying matrices $W^k$—similar to Algorithm 1.

**Setting.** We compare the following three topologies (and the corresponding matrices $W^k$):
• **Full.** Full graph at the end of each epoch, otherwise local steps. This means that we make 120 communication rounds (by communication round we mean the exchange of information between a pair of devices) in an epoch.
• **Local.** Full graph at the end of each 5th epoch, otherwise local steps. This means that we make 24 communication rounds in an epoch (in average: 4 epochs without communications and 1 epoch with 120 rounds).
• **Clusters.** At the end of each epoch, clique clusters of size 4 are randomly formed (in total 4 cliques). This means that we make 24 communication rounds in an epoch.

Note that the communication budget of the first approach is 5 times larger.

We use the same learning rate equal to 0.002 for the generator and discriminator. The rest of the parameters and features of the architecture can be found in the supplementary material.

**Results.** The results of the experiment are presented in Figure 3 and Figure 6 (Section A.3). In terms of the number of local epochs, all the methods converged quite close to each other and produced similar images. In terms of communications, Local and Cluster topologies lead to much better results, and the Cluster topology is slightly better than the Local.

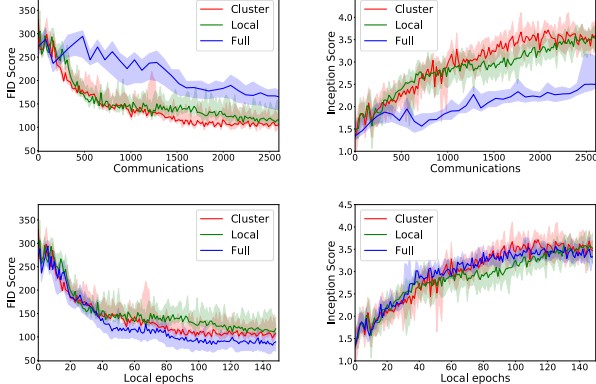

Figure 3: Comparison of the three network topologies in DCGAN distributed decentralized learning on CIFAR-10. FID Score and Inception Score vs the number of communications (two top), and the same Scores vs the number of local epochs (two bottom). The experiment was repeated 5 times on different random data splitting, the maximum and minimum deviations are depicted in the plots by the shade.

## 6 Conclusion

We propose a novel efficient algorithm for solving decentralized stochastic MVIs and SPPs under a very general assumption on the network topology and communication constraints. In particular, our method is the first decentralized extragradient method with local steps for time-varying network topologies. Moreover, for the proposed algorithm, we prove the convergence rate theorem in the SM, M and NM cases. In the numerical experiments, we verify that the dependence of our rates on the data heterogeneity parameter $D$ is tight in the SM case, and cannot be further improved in general. By training DCGAN on a decentralized topology, we demonstrate that our method is effective on practical DL tasks. As a future work it would be interesting to generalize such algorithms for infinite-dimensional problems.

## Acknowledgments

This research of A. Beznosikov has been supported by The Analytical Center for the Government of the Russian Federation (Agreement No. 70-2021-00143 dd. 01.11.2021, IGK 000000D730321P5Q0002). The work by P. Dvurechensky in Section C.3 was funded by the Deutsche Forschungsgemeinschaft (DFG, German Research Foundation) under Germany's Excellence Strategy – The Berlin Mathematics Research Center MATH+ (EXC-2046/1, project ID: 390685689).

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
