# Supplementary Material

## A Experiments

In this section, we provide additional details about the experiments reported in the main text and additional experiments with a decreasing stepsize. We implement all the methods using Python 3.8 using PyTorch [77] and Ray [68] and run the experiments on a machine with 24 AMD EPYC 7552 @ 2.20GHz processors, 2 GPUs NVIDIA A100-PCIE with 40536 Mb of memory each (Cuda 11.3).

### A.1 Additional Details of the Experiments with Training GANs

As mentioned in the main text of the paper, we use DCGAN architecture [79], conditioned by the class labels, similarly to [65]. The illustration of the architecture is provided in Figure 4. In Table 2, we give the hyperparameters for all the experiments.

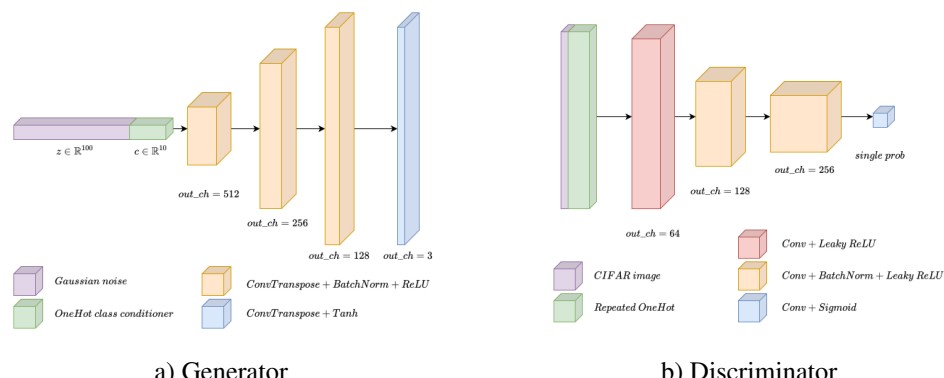

a) Generator            b) Discriminator

Figure 4: DCGAN architecture.

| Hyperparameters | |
| --- | --- |
| Batch size | =64 |
| Weight clipping for the discriminator | =0.01 |
| Learning rate for generator and discriminator | =0.002 |
| Initialization: | normal |
| Other parameters: | default in PyTorch |

Table 2: Hyperparameters for DCGAN training.

Next, we comment on the relation between the main assumptions of the theoretical analysis and the example of training GANs. First of all, the goal of the DCGAN training experiment is to study how the network topology influences the convergence of the algorithm. Even if the assumptions do not hold, we see that the algorithm performs quite well and is flexible w.r.t. the choice of the topology. Secondly, as we write in the main text, the made assumptions are quite standard and widely used in the literature. In particular, Assumptions 3.1 and 3.3 are classical and are often used in the literature, see, e.g. [37], including the literature on the neural networks training. Assumption 3.4 is also widely used [40, 96, 30, 85, 4, 31, 20, 32, 58], and holds with a small constant $D$ when the data is uniformly split among the devices. Such splitting can be easily made when one uses a computational cluster with a large amount of data, e.g., images. In Section 5.2, we deliberately consider a more difficult setup and make the distribution of images over the nodes not uniform, but heterogeneous. As we see, the results of the experiments are quite promising in this case. Assumption 3.2 (NM) is also used in the literature on the algorithms for training GANs and their analysis [58, 63, 57]. Moreover, this assumption is shown to hold in some nonconvex minimization problems, for example, when SGD is used for training neural networks [53, 43, 100].

## A.2 Additional Experiments with Decreasing Stepsize

As it can be seen from the proofs in the next sections, our theoretical results in Theorem 4.1 hold for the fixed stepsizes that optimize the error of the obtained approximate solution given the budget of $K$ iterations. Thus, given a target accuracy $\epsilon > 0$ and using the bounds in Theorem 4.1, we can choose the number of iterations $K = K(\epsilon)$ to guarantee the accuracy $\epsilon$. In turn, based on the value $K(\epsilon)$, we choose the fixed stepsize $\gamma = \gamma(K(\epsilon))$. This procedure of defining the stepsize based on the target accuracy and the corresponding budget of iterations is quite standard in the literature, see, e.g., [37, 88, 10]. In Section D, we provide a generic technique that allows us not to fix the target accuracy in advance and construct a decreasing sequence of stepsizes. This is useful when the desired target accuracy $\epsilon$ is not known, or not determined. In this section, we numerically illustrate that Algorithm 1 can reach arbitrarily small error when implemented with quite simple decreasing stepsizes. We note that, despite not analyzed theoretically, the used in the experiments decreasing stepsize leads to a good performance of the algorithm, which additionally illustrates the flexibility of our approach for practical purposes.

For the experiments, we consider the same setup as in Section 5.1 of the main text (see Figure 1, left), i.e., strongly-monotone bilinear objective functions distributed over the network with the ring topology. We consider two cases: with and without stochastic noise, i.e., we fix either $\sigma = 0$, or $\sigma = 100$. During the training, we decrease the stepsize as $\gamma_k = \frac{\alpha}{k+\beta}$, where $k$ is the current iteration number. We set $\alpha = 40, \beta = 800$ in the noiseless case and $\alpha = 15, \beta = 150$ when $\sigma^2 = 100$. In Figure 5, we can see that the error decreases to zero with a sublinear rate. This is in contrast to the limiting behavior which we observe in Figure 1, when the algorithm is not able to optimize below a certain threshold. Sublinear convergence may be explained by the second and third terms in the estimate (6).

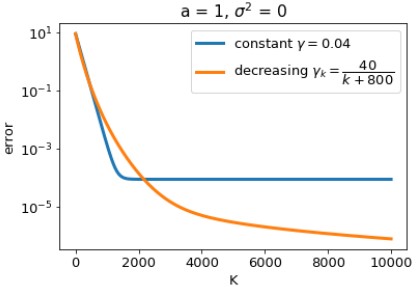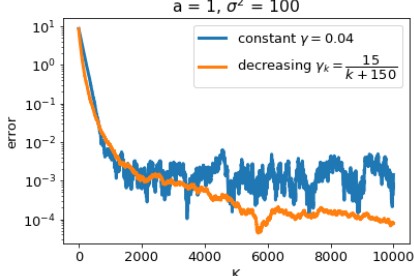

Figure 5: Convergence of Algorithm 1 with the decreasing stepsizes in the noiseless (left) and stochastic (right) cases.

## A.3 Images Generated by the Trained GAN

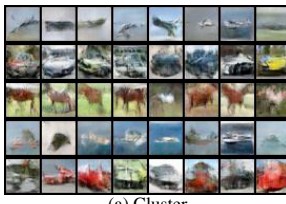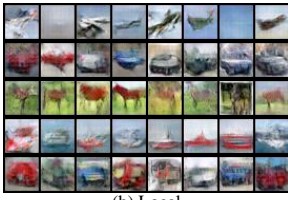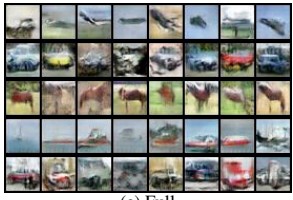

(a) Cluster                          (b) Local                          (c) Full

Figure 6: Images generated by DCGAN trained distributedly using different communication graph topologies: (a) Cluster, (b) Local, (c) Full.

# B Useful Facts Used in the Proofs

Before we start with the proofs, we give several simple facts that are used throughout the proofs of the main theorem.

**Upper bound for a squared sum.** For an arbitrary integer $n \geq 1$ and arbitrary set of vectors $a_1, \ldots, a_n$, we have

$$\left\| \sum_{i=1}^{n} a_i \right\|^2 \leq n \sum_{i=1}^{n} \|a_i\|^2. \tag{11}$$

**Cauchy-Schwarz inequality.** For arbitrary vectors $a$ and $b$ and any constant $c > 0$, we have

$$2\langle a, b \rangle \leq c\|a\|^2 + c^{-1}\|b\|^2, \tag{12}$$

$$\|a + b\|^2 \leq (1 + c)\|a\|^2 + (1 + c^{-1})\|b\|^2. \tag{13}$$

**Cauchy-Schwarz inequality for random variables.** Let $\xi$ and $\eta$ be real-valued random variables such that $\mathbb{E}[\xi^2] < \infty$ and $\mathbb{E}[\eta^2] < \infty$. Then

$$\mathbb{E}[\xi \eta] \leq \sqrt{\mathbb{E}[\xi^2]\mathbb{E}[\eta^2]}. \tag{14}$$

**Frobenius norm of product.** For given matrices $A$ and $B$, it holds that

$$\|AB\|_F \leq \|A\|_F \|B\|_2, \tag{15}$$

where $\|\cdot\|_F$ denotes the Frobenius norm of a matrix, and $\|\cdot\|_2$ is the spectral norm of a matrix, i.e., the maximal singular value.

# C  Missing Proofs for the Main Theorem

In this section, we provide the proof of the main theorem. For convenience, we give its full statement, including the explicit expressions for the stepsizes.

**Theorem C.1** (Theorem 4.1). *Let Assumptions 2.2, 3.1, 3.3, 3.4 hold and the sequences $\bar{z}^k$, $\widehat{z}^k$ be generated by Algorithm 1 that is run for $K > 0$ iterations. Then,*

- ***Strongly-monotone case:*** *under Assumption 3.2(SM), with* $\gamma = \min\left\{ \frac{p}{120\tau L}, \frac{2\ln\left(\max\{2, \mu^2 M r_0 K/(40\sigma^2)\}\right)}{\mu K} \right\}$ *it holds that*

$$\mathbb{E}\left[\|\bar{z}^{K+1} - z^*\|^2\right] = \tilde{\mathcal{O}}\left( \|z^0 - z^*\|^2 \cdot \exp\left( -\frac{\mu K p}{240 L \tau} \right) + \frac{\sigma^2}{\mu^2 M K} + \frac{L^2 \Delta}{\mu^4 K^2} \right);$$

- ***Monotone case:*** *under Assumption 3.2(M), for any convex compact $\mathcal{C}$ s.t. $z^0, z^* \in \mathcal{C}$ and $\max_{z, z' \in \mathcal{C}} \|z - z'\| \leq \Omega_{\mathcal{C}}$, with $\gamma = \min\left\{ \frac{1}{3L}, \left( \frac{2\Omega_{\mathcal{C}}^2 M}{5(K+1)\sigma^2} \right)^{\frac{1}{2}}, \left( \frac{\Omega_{\mathcal{C}}^2}{6(K+1)^2 L^2 \Delta} \right)^{\frac{1}{4}} \right\}$ it holds that*

$$\sup_{z \in \mathcal{C}} \mathbb{E}\left[\langle F(z), \widehat{z}^K - z \rangle\right] = \mathcal{O}\left( \frac{L\Omega_{\mathcal{C}}^2}{K} + \frac{\sigma \Omega_{\mathcal{C}}}{\sqrt{MK}} + \frac{\sqrt{L\Omega_{\mathcal{C}}^3 \sqrt{\Delta}}}{\sqrt{K}} + \sqrt{\frac{(\Delta + L^2 \Omega_{\mathcal{C}}^2)\Omega_{\mathcal{C}} \sqrt{\Delta}}{KL}} \right).$$

*Under the additional assumption that, for all $k$, $\|\bar{z}^k\| \leq \Omega$, with $\gamma = \min\left\{ \frac{1}{3L}, \left( \frac{\Omega_{\mathcal{C}}^2 M}{20(K+1)\sigma^2} \right)^{\frac{1}{2}}, \left( \frac{\Omega_{\mathcal{C}}^2}{60(K+1)^2 L^2 \Delta} \right)^{\frac{1}{4}}, \left( \frac{\Omega_{\mathcal{C}}^2}{(K+1)((\Omega + \Omega_{\mathcal{C}})L\sqrt{\Delta} + \Delta)} \right)^{\frac{1}{2}} \right\}$, we have that*

$$\sup_{z \in \mathcal{C}} \mathbb{E}\left[\langle F(z), \widehat{z}^K - z \rangle\right] = \mathcal{O}\left( \frac{L\Omega_{\mathcal{C}}^2}{K} + \frac{\sigma \Omega_{\mathcal{C}}}{\sqrt{MK}} + \frac{\sqrt{L\Omega_{\mathcal{C}}^3 \sqrt{\Delta}}}{K^{3/4}} + \sqrt{\frac{((\Omega + \Omega_{\mathcal{C}})L\sqrt{\Delta} + \Delta)\Omega_{\mathcal{C}}^2}{K}} \right);$$

- ***Non-monotone case:*** *under Assumption 3.2(NM) and if $\|z^0\| \leq \Omega, \|z^*\| \leq \Omega$, with $\gamma = \min\left\{ \frac{1}{5L}, \left( \frac{\|z^0 - z^*\|^2}{(K+1)^2 L^2 \Delta} \right)^{1/4} \right\}$:*

$$\mathbb{E}\left[ \frac{1}{K+1} \sum_{k=0}^{K} \|F(\bar{z}^k)\|^2 \right] = \mathcal{O}\left( \frac{L^2 \Omega^2}{K} + \frac{\sigma^2}{M} + L\Omega\sqrt{\Delta} + \frac{\sqrt{L\Omega\Delta^{3/4}}}{\sqrt{K}} \right).$$

*Under an additional assumption that, for all $k$, $\|\bar{z}^k\| \leq \Omega$, with $\gamma = \min\left\{\frac{1}{5L}, \left(\frac{\Omega^2}{(K+1)L\Delta}\right)^{1/3}\right\}$,*

*we have that*

$$\mathbb{E}\left[\frac{1}{K+1}\sum_{k=0}^{K}\|F(\bar{z}^k)\|^2\right] = \mathcal{O}\left(\frac{L^2\Omega^2}{K} + \frac{\sigma^2}{M} + \frac{(L\Omega\Delta)^{2/3}}{K^{1/3}} + L\Omega\sqrt{\Delta}\right).$$

*Here $\Delta = \frac{\tau}{p}\left(\frac{D^2\tau}{p} + \sigma^2\right)$.*

We start with some convenient notations, and then proceed with the proof on the case-by-case basis: strongly-monotone, monotone, non-monotone under the Minty condition.

## C.1 Notation

We introduce some auxiliary notation as follows.

• Average across all the devices/nodes values of the iterates $z$ and the stochastic realizations of the operators $g$ at iteration $k$:

$$\bar{z}^k := \frac{1}{M}\sum_{m=1}^{M}z_m^k, \quad \bar{g}^k := \frac{1}{M}\sum_{m=1}^{M}g_m^k = \frac{1}{M}\sum_{m=1}^{M}F_m(z_m^k, \xi_m^k), \tag{16}$$

$$\bar{z}^{k+1/3} := \frac{1}{M}\sum_{m=1}^{M}z_m^{k+1/3}, \quad \bar{g}^{k+1/3} := \frac{1}{M}\sum_{m=1}^{M}g_m^{k+1/3} = \frac{1}{M}\sum_{m=1}^{M}F_m(z_m^{k+1/3}, \xi_m^{k+1/3}), \tag{17}$$

$$\bar{z}^{k+1/3} = \bar{z}^k - \gamma\bar{g}^k, \quad \bar{z}^{k+2/3} = \bar{z}^k - \gamma\bar{g}^{k+1/3}, \quad \bar{z}^{k+1} = \bar{z}^{k+2/3}. \tag{18}$$

The last equality with $\bar{z}^{k+1} = \bar{z}^{k+2/3}$ follows from the fact that one step of the gossip procedure, i.e., step 7 of Algorithm 1 preserves the average over $m$ since the matrix $W$ is doubly stochastic (see Definition 2.1).

• Matrix notation for the collection over all the nodes of the iterates $z$, of the averaged iterates $\bar{z}$, of the stochastic realizations of the operators $g$ and of the averaged stochastic realizations of the operators $\bar{g}$ at iteration $k$:

$$Z^k := [z_1^k, \ldots, z_M^k], \quad \bar{Z}^k := [\bar{z}^k, \ldots, \bar{z}^k], \tag{19}$$

$$G^k := [g_1^k, \ldots, g_M^k], \quad \bar{G}^k := [\bar{g}^k, \ldots, \bar{g}^k], \tag{20}$$

$$\Phi^k := [F_1(z_1^k), \ldots, F_M(z_M^k)], \quad \bar{\Phi}^k := \left[\frac{1}{M}\sum_{m=1}^{M}F_m(z_m^k), \ldots, \frac{1}{M}\sum_{m=1}^{M}F_m(z_m^k)\right]. \tag{21}$$

This notation allows us to rewrite compactly the iterations of Algorithm 1 and the "averaged" dynamics given in (18):

$$Z^{k+1/3} = Z^k - \gamma G^k, \quad \bar{Z}^{k+1/3} = \bar{Z}^k - \gamma\bar{G}^k,$$

$$Z^{k+2/3} = Z^k - \gamma G^{k+1/3}, \quad \bar{Z}^{k+2/3} = \bar{Z}^k - \gamma\bar{G}^{k+1/3}, \tag{22}$$

$$Z^{k+1} = Z^{k+2/3}W^k, \quad \bar{Z}^{k+1} = \bar{Z}^{k+2/3}.$$

• Average over devices consensus errors

$$\text{Err}(k) = \frac{1}{M}\sum_{m=1}^{M}\|z_m^k - \bar{z}^k\|^2, \quad \text{Err}(k+1/3) = \frac{1}{M}\sum_{m=1}^{M}\|z_m^{k+1/3} - \bar{z}^{k+1/3}\|^2, \tag{23}$$

which can be interpreted as the measure of the discrepancy in the values of the current iterate between the devices at iteration $k$.

## C.2 Proof of Theorem 4.1, Strongly-Monotone Case.

### C.2.1 General Per-Iterate Estimate

The goal of this subsection is to derive a general bound for the per-iteration progress of Algorithm 1. This bound will be further refined in the following subsections. We begin the proof with the following lemma.

**Lemma C.2.** *Let $z, y \in \mathbb{R}^n$. Defining $z^+ = z - y$, for all $u \in \mathbb{R}^n$, we have*

$$\|z^+ - u\|^2 = \|z - u\|^2 - 2\langle y, z^+ - u\rangle - \|z^+ - z\|^2.$$

**Proof:** Simple calculations give the following chain of equalities.

$$
\begin{aligned}
\|z^+ - u\|^2 &= \|z^+ - z + z - u\|^2 \\
&= \|z - u\|^2 + 2\langle z^+ - z, z - u\rangle + \|z^+ - z\|^2 \\
&= \|z - u\|^2 + 2\langle z^+ - z, z^+ - u\rangle - \|z^+ - z\|^2 \\
&= \|z - u\|^2 + 2\langle z^+ - (z - y), z^+ - u\rangle - 2\langle y, z^+ - u\rangle - \|z^+ - z\|^2 \\
&= \|z - u\|^2 - 2\langle y, z^+ - u\rangle - \|z^+ - z\|^2.
\end{aligned}
$$

$\square$

Applying the previous lemma with $z^+ = \bar{z}^{k+2/3}$, $z = \bar{z}^k$, $u = z^*$ and $y = \gamma\bar{g}^{k+1/3}$, we have

$$\|\bar{z}^{k+2/3} - z^*\|^2 = \|\bar{z}^k - z^*\|^2 - 2\gamma\langle\bar{g}^{k+1/3}, \bar{z}^{k+2/3} - z^*\rangle - \|\bar{z}^{k+2/3} - \bar{z}^k\|^2.$$

From the same lemma, but with $z^+ = \bar{z}^{k+1/3}$, $z = \bar{z}^k$, $u = z^{k+2/3}$, $y = \gamma\bar{g}^k$, we have

$$\|\bar{z}^{k+1/3} - \bar{z}^{k+2/3}\|^2 = \|\bar{z}^k - \bar{z}^{k+2/3}\|^2 - 2\gamma\langle\bar{g}^k, \bar{z}^{k+1/3} - \bar{z}^{k+2/3}\rangle - \|\bar{z}^{k+1/3} - \bar{z}^k\|^2.$$

Combining the two previous equalities, we obtain

$$
\begin{aligned}
\|\bar{z}^{k+2/3} - z^*\|^2 + \|\bar{z}^{k+1/3} - \bar{z}^{k+2/3}\|^2 = & \|\bar{z}^k - z^*\|^2 - \|\bar{z}^{k+1/3} - \bar{z}^k\|^2 \\
& - 2\gamma\langle\bar{g}^{k+1/3}, \bar{z}^{k+2/3} - z^*\rangle - 2\gamma\langle\bar{g}^k, \bar{z}^{k+1/3} - \bar{z}^{k+2/3}\rangle.
\end{aligned}
$$

A small rearrangement of the terms gives

$$
\begin{aligned}
\|\bar{z}^{k+2/3} &- z^*\|^2 + \|\bar{z}^{k+1/3} - \bar{z}^{k+2/3}\|^2 \\
&= \|\bar{z}^k - z^*\|^2 - \|\bar{z}^{k+1/3} - \bar{z}^k\|^2 \\
&\quad - 2\gamma\langle\bar{g}^{k+1/3}, \bar{z}^{k+1/3} - z^*\rangle + 2\gamma\langle\bar{g}^{k+1/3} - \bar{g}^k, \bar{z}^{k+1/3} - \bar{z}^{k+2/3}\rangle \\
&\overset{(12)}{\leq} \|\bar{z}^k - z^*\|^2 - \|\bar{z}^{k+1/3} - \bar{z}^k\|^2 \\
&\quad - 2\gamma\langle\bar{g}^{k+1/3}, \bar{z}^{k+1/3} - z^*\rangle + \gamma^2\|\bar{g}^{k+1/3} - \bar{g}^k\|^2 + \|\bar{z}^{k+1/3} - \bar{z}^{k+2/3}\|^2.
\end{aligned}
$$

Taking the full expectation, we get

$$
\begin{aligned}
\mathbb{E}\left[\|\bar{z}^{k+2/3} - z^*\|^2\right] \leq \mathbb{E}\left[\|\bar{z}^k - z^*\|^2\right] - \mathbb{E}\left[\|\bar{z}^{k+1/3} - \bar{z}^k\|^2\right] \\
- 2\gamma\mathbb{E}\left[\langle\bar{g}^{k+1/3}, \bar{z}^{k+1/3} - z^*\rangle\right] + \gamma^2\mathbb{E}\left[\|\bar{g}^{k+1/3} - \bar{g}^k\|^2\right].
\end{aligned}
$$

Since $\bar{z}^{k+1} = \bar{z}^{k+2/3}$, we obtain the following general bound for the per-iteration progress of Algorithm 1:

$$
\begin{aligned}
\mathbb{E}\left[\|\bar{z}^{k+1} - z^*\|^2\right] \leq \mathbb{E}\left[\|\bar{z}^k - z^*\|^2\right] - \mathbb{E}\left[\|\bar{z}^{k+1/3} - \bar{z}^k\|^2\right] \\
- 2\gamma\mathbb{E}\left[\langle\bar{g}^{k+1/3}, \bar{z}^{k+1/3} - z^*\rangle\right] + \gamma^2\mathbb{E}\left[\|\bar{g}^{k+1/3} - \bar{g}^k\|^2\right]. \quad (24)
\end{aligned}
$$

Our next goal is to consider in more details the last two terms, i.e., $-2\gamma\mathbb{E}\left[\langle\bar{g}^{k+1/3}, \bar{z}^{k+1/3} - z^*\rangle\right]$ and $\gamma^2\mathbb{E}\left[\|\bar{g}^{k+1/3} - \bar{g}^k\|^2\right]$, and estimate them from above.

### C.2.2 Two Auxiliary Estimates for the General Bound

In the following two auxiliary lemmas, we prove the estimates mentioned in the end of the previous subsection.

**Lemma C.3.** *Under Assumptions 3.1, 3.2(SM), 3.3 it holds that*

$$-2\gamma\mathbb{E}\left[\langle\bar{g}^{k+1/3}, \bar{z}^{k+1/3} - z^*\rangle\right] \leq -\gamma\mu\mathbb{E}\left[\|\bar{z}^{k+1/3} - z^*\|^2\right] + \frac{\gamma L^2}{\mu}\mathbb{E}\left[\operatorname{Err}(k + 1/3)\right]. \quad (25)$$

**Proof:** First, we use the independence of all random vectors $\xi^i = (\xi_1^i, \ldots, \xi_m^i)$ and take the conditional expectation $\mathbb{E}_{\xi^{k+1/3}}$ w.r.t. the vector $\xi^{k+1/3}$, conditioned on the other randomness. This gives:

$$-2\gamma\mathbb{E}\left[\langle \bar{g}^{k+1/3}, \bar{z}^{k+1/3} - z^* \rangle\right] \overset{(17)}{=} -2\gamma\mathbb{E}\left[\left\langle \frac{1}{M}\sum_{m=1}^{M}\mathbb{E}_{\xi^{k+1/3}}[F_m(z_m^{k+1/3}, \xi_m^{k+1/3})], \bar{z}^{k+1/3} - z^* \right\rangle\right]$$

$$\overset{(4)}{=} -2\gamma\mathbb{E}\left[\left\langle \frac{1}{M}\sum_{m=1}^{M}F_m(z_m^{k+1/3}), \bar{z}^{k+1/3} - z^* \right\rangle\right]$$

$$= -2\gamma\mathbb{E}\left[\left\langle \frac{1}{M}\sum_{m=1}^{M}F_m(\bar{z}^{k+1/3}), \bar{z}^{k+1/3} - z^* \right\rangle\right]$$

$$+ 2\gamma\mathbb{E}\left[\left\langle \frac{1}{M}\sum_{m=1}^{M}[F_m(\bar{z}^{k+1/3}) - F_m(z_m^{k+1/3})], \bar{z}^{k+1/3} - z^* \right\rangle\right]$$

$$= -2\gamma\mathbb{E}\left[\left\langle F(\bar{z}^{k+1/3}), \bar{z}^{k+1/3} - z^* \right\rangle\right]$$

$$+ 2\gamma\mathbb{E}\left[\left\langle \frac{1}{M}\sum_{m=1}^{M}[F_m(\bar{z}^{k+1/3}) - F_m(z_m^{k+1/3})], \bar{z}^{k+1/3} - z^* \right\rangle\right]. \tag{26}$$

Next, we prove that $\langle F(z^*), \bar{z}^{k+1/3} - z^* \rangle \geq 0$ by contradiction. To that end, assume that $\langle F(z^*), \bar{z}^{k+1/3} - z^* \rangle < 0$. By Assumption 3.1, $F$ is $L$-Lipschitz continuous, and, hence, for a small enough $\alpha > 0$ it holds that $\langle F(\hat{z}), \bar{z}^{k+1/3} - z^* \rangle < 0$, where $\hat{z} = z^* + \alpha(\bar{z}^{k+1/3} - z^*)$. We substitute $\alpha\bar{z}^{k+1/3} = \hat{z} - (1 - \alpha)z^*$ to the inequality $\langle F(\hat{z}), \alpha\bar{z}^{k+1/3} - \alpha z^* \rangle < 0$ and get $\langle F(\hat{z}), \hat{z} - z^* \rangle < 0$. But, this contradicts the definition of the solution $z^*$ in (1) which implies that $\langle F(\hat{z}), \hat{z} - z^* \rangle \geq 0$. Thus, we have that $\langle F(z^*), \bar{z}^{k+1/3} - z^* \rangle \geq 0$. We combine this inequality with (26) and obtain

$$-2\gamma\mathbb{E}\left[\langle \bar{g}^{k+1/3}, \bar{z}^{k+1/3} - z^* \rangle\right] \overset{(26)}{\leq} -2\gamma\mathbb{E}\left[\left\langle F(\bar{z}^{k+1/3}) - F(z^*), \bar{z}^{k+1/3} - z^* \right\rangle\right]$$

$$+ 2\gamma\mathbb{E}\left[\left\langle \frac{1}{M}\sum_{m=1}^{M}[F_m(\bar{z}^{k+1/3}) - F_m(z_m^{k+1/3})], \bar{z}^{k+1/3} - z^* \right\rangle\right]$$

$$\overset{(SM)}{\leq} -2\gamma\mu\mathbb{E}\left[\|\bar{z}^{k+1/3} - z^*\|^2\right]$$

$$+ 2\gamma\mathbb{E}\left[\left\langle \frac{1}{M}\sum_{m=1}^{M}[F_m(\bar{z}^{k+1/3}) - F_m(z_m^{k+1/3})], \bar{z}^{k+1/3} - z^* \right\rangle\right].$$

Applying (12) with $c = \mu > 0$, we further get

$$-2\gamma\mathbb{E}\left[\langle \bar{g}^{k+1/3}, \bar{z}^{k+1/3} - z^* \rangle\right] \leq -2\gamma\mu\mathbb{E}\left[\|\bar{z}^{k+1/3} - z^*\|^2\right]$$

$$+ \gamma\mu\mathbb{E}\left[\left\|\bar{z}^{k+1/3} - z^*\right\|^2\right] + \frac{\gamma}{\mu}\mathbb{E}\left[\left\|\frac{1}{M}\sum_{m=1}^{M}[F_m(\bar{z}^{k+1/3}) - F_m(z_m^{k+1/3})]\right\|^2\right]$$

$$= -\gamma\mu\mathbb{E}\left[\|\bar{z}^{k+1/3} - z^*\|^2\right] + \frac{\gamma}{\mu M^2}\mathbb{E}\left[\left\|\sum_{m=1}^{M}[F_m(\bar{z}^{k+1/3}) - F_m(z_m^{k+1/3})]\right\|^2\right]$$

$$\overset{(11)}{\leq} -\gamma\mu\mathbb{E}\left[\|\bar{z}^{k+1/3} - z^*\|^2\right] + \frac{\gamma}{\mu M}\mathbb{E}\left[\sum_{m=1}^{M}\left\|F_m(\bar{z}^{k+1/3}) - F_m(z_m^{k+1/3})\right\|^2\right]$$

$$\overset{(L)}{\leq} -\gamma\mu\mathbb{E}\left[\|\bar{z}^{k+1/3} - z^*\|^2\right] + \frac{\gamma L^2}{\mu M}\mathbb{E}\left[\sum_{m=1}^{M}\left\|\bar{z}^{k+1/3} - z_m^{k+1/3}\right\|^2\right].$$

Applying (23) to the last term, we finish the proof.

$\square$

**Lemma C.4.** *Under Assumptions 3.1, 3.3 it holds that*

$$\mathbb{E}\left[\|\bar{g}^{k+1/3} - \bar{g}^k\|^2\right] \leq 5L^2\mathbb{E}\left[\|\bar{z}^{k+1/3} - \bar{z}^k\|^2\right] + \frac{10\sigma^2}{M}$$
$$+ 5L^2\mathbb{E}\left[\text{Err}(k+1/3)\right] + 5L^2\mathbb{E}\left[\text{Err}(k)\right]. \qquad (27)$$

**Proof:** Consider the following chain of inequalities:

$$\mathbb{E}\left[\|\bar{g}^{k+1/3} - \bar{g}^k\|^2\right] \overset{(16),(17)}{=} \mathbb{E}\left[\left\|\frac{1}{M}\sum_{m=1}^M F_m(z_m^{k+1/3}, \xi_m^{k+1/3}) - \frac{1}{M}\sum_{m=1}^M F_m(z_m^k, \xi_m^k)\right\|^2\right]$$

$$\leq 5\mathbb{E}\left[\left\|\frac{1}{M}\sum_{m=1}^M [F_m(z_m^{k+1/3}, \xi_m^{k+1/3}) - F_m(z_m^{k+1/3})]\right\|^2\right]$$

$$+ 5\mathbb{E}\left[\left\|\frac{1}{M}\sum_{m=1}^M [F_m(z_m^{k+1/3}) - F_m(\bar{z}^{k+1/3})]\right\|^2\right] + 5\mathbb{E}\left[\left\|\frac{1}{M}\sum_{m=1}^M [F_m(\bar{z}^{k+1/3}) - F_m(\bar{z}^k)]\right\|^2\right]$$

$$+ 5\mathbb{E}\left[\left\|\frac{1}{M}\sum_{m=1}^M [F_m(z_m^k) - F_m(\bar{z}^k)]\right\|^2\right] + 5\mathbb{E}\left[\left\|\frac{1}{M}\sum_{m=1}^M [F_m(z_m^k, \xi_m^k) - F_m(z_m^k)]\right\|^2\right]$$

$$\overset{(11)}{\leq} 5\mathbb{E}\left[\left\|\frac{1}{M}\sum_{m=1}^M [F_m(z_m^{k+1/3}, \xi_m^{k+1/3}) - F_m(z_m^{k+1/3})]\right\|^2\right]$$

$$+ 5\mathbb{E}\left[\left\|\frac{1}{M}\sum_{m=1}^M [F_m(z_m^k, \xi_m^k) - F_m(z_m^k)]\right\|^2\right]$$

$$+ \frac{5}{M}\sum_{m=1}^M \mathbb{E}\left[\left\|F_m(z_m^{k+1/3}) - F_m(\bar{z}^{k+1/3})\right\|^2\right] + \frac{5}{M}\sum_{m=1}^M \mathbb{E}\left[\left\|F_m(z_m^k) - F_m(\bar{z}^k)\right\|^2\right]$$

$$+ 5\mathbb{E}\left[\left\|F(\bar{z}^{k+1/3}) - F(\bar{z}^k)\right\|^2\right]$$

$$\overset{(L),(23)}{\leq} 5\mathbb{E}\left[\left\|\frac{1}{M}\sum_{m=1}^M [F_m(z_m^{k+1/3}, \xi_m^{k+1/3}) - F_m(z_m^{k+1/3})]\right\|^2\right]$$

$$+ 5\mathbb{E}\left[\left\|\frac{1}{M}\sum_{m=1}^M [F_m(z_m^k, \xi_m^k) - F_m(z_m^k)]\right\|^2\right]$$

$$+ 5L^2\mathbb{E}\left[\text{Err}(k+1/3)\right] + 5L^2\mathbb{E}\left[\text{Err}(k)\right] + 5L^2\mathbb{E}\left[\|\bar{z}^{k+1/3} - \bar{z}^k\|^2\right]$$

$$= 5\mathbb{E}\left[\mathbb{E}_{\xi^{k+1/3}}\left[\left\|\frac{1}{M}\sum_{m=1}^M [F_m(z_m^{k+1/3}, \xi_m^{k+1/3}) - F_m(z_m^{k+1/3})]\right\|^2\right]\right]$$

$$+ 5\mathbb{E}\left[\mathbb{E}_{\xi^k}\left[\left\|\frac{1}{M}\sum_{m=1}^M [F_m(z_m^k, \xi_m^k) - F_m(z_m^k)]\right\|^2\right]\right]$$

$$+ 5L^2\mathbb{E}\left[\text{Err}(k+1/3)\right] + 5L^2\mathbb{E}\left[\text{Err}(k)\right] + 5L^2\mathbb{E}\left[\|\bar{z}^{k+1/3} - \bar{z}^k\|^2\right].$$

Using the independence of the realizations of $\xi^k, \xi^{k+1/3}$ in each node and (4), we get:

$$\mathbb{E}\left[\|\bar{g}^{k+1/3} - \bar{g}^k\|^2\right] \leq \frac{10\sigma^2}{M} + 5L^2\mathbb{E}\left[\text{Err}(k+1/3)\right] + 5L^2\mathbb{E}\left[\text{Err}(k)\right] + 5L^2\mathbb{E}\left[\|\bar{z}^{k+1/3} - \bar{z}^k\|^2\right].$$

$\square$

### C.2.3 Refined General Bound

We now return to the general bound for the per-iteration progress and combine the general bound (24) with the two estimates (25) and (27) obtained in the previous subsection. In this way, we obtain

$$
\mathbb{E}\left[\|\bar{z}^{k+1} - z^*\|^2\right] \leq \mathbb{E}\left[\|\bar{z}^k - z^*\|^2\right] - \mathbb{E}\left[\|\bar{z}^{k+1/3} - \bar{z}^k\|^2\right]
$$

$$
- \gamma\mu\mathbb{E}\left[\|\bar{z}^{k+1/3} - z^*\|^2\right] + \frac{\gamma L^2}{\mu}\mathbb{E}\left[\mathrm{Err}(k+1/3)\right]
$$

$$
+ \gamma^2\left(5L^2\mathbb{E}\left[\|\bar{z}^{k+1/3} - \bar{z}^k\|^2\right] + \frac{10\sigma^2}{M} + 5L^2\mathbb{E}\left[\mathrm{Err}(k+1/3)\right] + 5L^2\mathbb{E}\left[\mathrm{Err}(k)\right]\right).
$$

Using (13) with $c = 1$, $a = \bar{z}^{k+1/3} - z^*$ and $b = \bar{z}^{k+1/3} - \bar{z}^k$, we get

$$
-\|\bar{z}^{k+1/3} - z^*\|^2 \leq -\tfrac{1}{2}\|\bar{z}^k - z^*\|^2 + \|\bar{z}^{k+1/3} - \bar{z}^k\|^2,
$$

which in combination with the previous inequality gives

$$
\mathbb{E}\left[\|\bar{z}^{k+1} - z^*\|^2\right] \leq \left(1 - \frac{\gamma\mu}{2}\right)\mathbb{E}\left[\|\bar{z}^k - z^*\|^2\right] - (1 - 5\gamma^2 L^2 - \gamma\mu)\mathbb{E}\left[\|\bar{z}^{k+1/3} - \bar{z}^k\|^2\right]
$$

$$
+ \left(\frac{\gamma L^2}{\mu} + 5\gamma^2 L^2\right)\mathbb{E}\left[\mathrm{Err}(k+1/3)\right] + 5\gamma^2 L^2\mathbb{E}\left[\mathrm{Err}(k)\right] + \frac{10\gamma^2\sigma^2}{M}. \tag{28}
$$

Since, by the Theorem assumptions, we have $\gamma \leq \frac{1}{3L}$, the refined general bound for the per-iteration progress becomes

$$
\mathbb{E}\left[\|\bar{z}^{k+1} - z^*\|^2\right] \leq \left(1 - \frac{\gamma\mu}{2}\right)\mathbb{E}\left[\|\bar{z}^k - z^*\|^2\right]
$$

$$
+ \left(\frac{\gamma L^2}{\mu} + 5\gamma^2 L^2\right)\mathbb{E}\left[\mathrm{Err}(k+1/3)\right] + 5\gamma^2 L^2\mathbb{E}\left[\mathrm{Err}(k)\right] + \frac{10\gamma^2\sigma^2}{M}. \tag{29}
$$

Our next goal is to bound the consensus error terms $\mathbb{E}\left[\mathrm{Err}(k)\right]$ and $\mathbb{E}\left[\mathrm{Err}(k+1/3)\right]$.

### C.2.4 Bounds for the Consensus Errors

The bounds for the consensus error terms are proved in the following two technical lemmas that give recursions for these error terms.

**Lemma C.5.** *Under Assumptions 3.1, 3.3, 3.4, 2.2, for $h = \lfloor k/\tau \rfloor - 1$, it holds that*

$$
\mathbb{E}\left[\mathrm{Err}(k)\right] \leq \left(1 - \frac{3p}{4}\right)\mathbb{E}[\mathrm{Err}(h\tau)] + \frac{144\gamma^2 L^2\tau}{p}\sum_{j=h\tau}^{k-1}\mathbb{E}\left[\mathrm{Err}(j+1/3)\right]
$$

$$
+ \left(\frac{72D^2\tau}{p} + 8\sigma^2\right)\sum_{j=h\tau}^{k-1}\gamma^2, \tag{30}
$$

$$
\mathbb{E}\left[\mathrm{Err}(k+1/3)\right] \leq \left(1 - \frac{3p}{4}\right)\mathbb{E}[\mathrm{Err}(h\tau)] + \frac{216\gamma^2 L^2\tau}{p}\sum_{j=h\tau}^{k-1}\mathbb{E}\left[\mathrm{Err}(j+1/3)\right] + \frac{216\gamma^2 L^2\tau}{p}\mathbb{E}\left[\mathrm{Err}(k)\right]
$$

$$
+ \left(\frac{108D^2\tau}{p} + 12\sigma^2\right)\sum_{j=h\tau}^{k-1}\gamma^2 + \left(\frac{108D^2\tau}{p} + 12\sigma^2\right)\gamma^2. \tag{31}
$$

**Proof:** Using the matrix notation introduced in (19), (20), (21), (22), we rewrite the error $\mathrm{Err}(k)$ as follows:

$$
M \cdot \mathbb{E}\left[\mathrm{Err}(k)\right] = \mathbb{E}\|Z^k - \bar{Z}^k\|_F^2 = \mathbb{E}\|Z^k - \bar{Z}^{h\tau} - \bar{Z}^k + \bar{Z}^{h\tau}\|_F^2
$$

$$
= \mathbb{E}\left[\left\|Z^{h\tau}\prod_{i=h\tau}^{k-1}W^i - \bar{Z}^{h\tau} - \gamma\sum_{j=h\tau}^{k-1}G^{j+1/3}\prod_{i=j}^{k-1}W^i\right.\right.
$$

$$- \left( \bar{Z}^{h\tau} \prod_{i=h\tau}^{k-1} W^i - \bar{Z}^{h\tau} - \gamma \sum_{j=h\tau}^{k-1} \bar{G}^{j+1/3} \prod_{i=j}^{k-1} W^i \right) \Bigg\|_F^2 \Bigg]$$

$$= \mathbb{E} \Bigg[ \mathbb{E}_{\xi^{(k-1)+1/3}} \Bigg[ \Bigg\| Z^{h\tau} \prod_{i=h\tau}^{k-1} W^i - \bar{Z}^{h\tau} - \left( \bar{Z}^{h\tau} \prod_{i=h\tau}^{k-1} W^i - \bar{Z}^{h\tau} \right) $$
$$- \gamma \sum_{j=h\tau}^{k-1} (\Phi^{j+1/3} - \bar{\Phi}^{j+1/3}) \prod_{i=j}^{k-1} W^i$$
$$- \gamma \sum_{j=h\tau}^{k-1} (G^{j+1/3} - \Phi^{j+1/3} - \bar{G}^{j+1/3} + \bar{\Phi}^{j+1/3}) \prod_{i=j}^{k-1} W^i \Bigg\|_F^2 \Bigg] \Bigg].$$

Since only $G^{(k-1)+1/3}$ and $\Phi^{(k-1)+1/3}$ depend on $\xi^{(k-1)+1/3}$, and $\mathbb{E}_{\xi^{(k-1)+1/3}} G^{(k-1)+1/3} = \Phi^{(k-1)+1/3}$, $\mathbb{E}_{\xi^{(k-1)+1/3}} \bar{G}^{(k-1)+1/3} = \bar{\Phi}^{(k-1)+1/3}$ (stochastic oracle is unbiased, see (4)), we have

$$M \cdot \mathbb{E}\left[ \mathrm{Err}(k) \right] = \mathbb{E} \Bigg[ \Bigg\| Z^{h\tau} \prod_{i=h\tau}^{k-1} W^i - \bar{Z}^{h\tau} - \left( \bar{Z}^{h\tau} \prod_{i=h\tau}^{k-1} W^i - \bar{Z}^{h\tau} \right) $$
$$- \gamma \sum_{j=h\tau}^{k-1} (\Phi^{j+1/3} - \bar{\Phi}^{j+1/3}) \prod_{i=j}^{k-1} W^i$$
$$- \gamma \sum_{j=h\tau}^{k-2} (G^{j+1/3} - \Phi^{j+1/3} - \bar{G}^{j+1/3} + \bar{\Phi}^{j+1/3}) \prod_{i=j}^{k-1} W^i \Bigg\|_F^2 \Bigg]$$
$$+ \gamma^2 \mathbb{E} \Bigg[ \Bigg\| (G^{(k-1)+1/3} - \Phi^{(k-1)+1/3} - \bar{G}^{(k-1)+1/3} + \bar{\Phi}^{(k-1)+1/3}) W^{k-1} \Bigg\|_F^2 \Bigg].$$

Next, we consider the blue term in the previous display and apply (13) with $c = \beta_1$, where the constant $\beta_1$ is defined below, $a = \Phi^{(k-1)+1/3} - \bar{\Phi}^{(k-1)+1/3}$ and $b$ collecting all the other terms in the first squared norm. This, combined with (15) and the fact that $\|W^{k-1}\|_2 \le 1$, gives us

$$M \cdot \mathbb{E}\left[ \mathrm{Err}(k) \right] \le (1 + \beta_1) \mathbb{E} \Bigg[ \Bigg\| Z^{h\tau} \prod_{i=h\tau}^{k-1} W^i - \bar{Z}^{h\tau} - \left( \bar{Z}^{h\tau} \prod_{i=h\tau}^{k-1} W^i - \bar{Z}^{h\tau} \right) $$
$$- \gamma \sum_{j=h\tau}^{k-2} (\Phi^{j+1/3} - \bar{\Phi}^{j+1/3}) \prod_{i=j}^{k-1} W^i$$
$$- \gamma \sum_{j=h\tau}^{k-2} (G^{j+1/3} - \Phi^{j+1/3} - \bar{G}^{j+1/3} + \bar{\Phi}^{j+1/3}) \prod_{i=j}^{k-1} W^i \Bigg\|_F^2 \Bigg]$$
$$+ (1 + \beta_1^{-1}) \gamma^2 \mathbb{E} \Bigg[ \Bigg\| \Phi^{(k-1)+1/3} - \bar{\Phi}^{(k-1)+1/3} \Bigg\|_F^2 \Bigg]$$
$$+ \gamma^2 \mathbb{E} \Bigg[ \Bigg\| G^{(k-1)+1/3} - \Phi^{(k-1)+1/3} - \bar{G}^{(k-1)+1/3} + \bar{\Phi}^{(k-1)+1/3} \Bigg\|_F^2 \Bigg].$$

In the same way, again using the unbiasedness, we separate the terms with index $(k-2) + 1/3$ using also (13) with $c = \beta_2$, where the constant $\beta_2$ is defined below

$$M \cdot \mathbb{E}\left[ \mathrm{Err}(k) \right] \le (1 + \beta_1) \mathbb{E} \Bigg[ \Bigg\| Z^{h\tau} \prod_{i=h\tau}^{k-1} W^i - \bar{Z}^{h\tau} - \left( \bar{Z}^{h\tau} \prod_{i=h\tau}^{k-1} W^i - \bar{Z}^{h\tau} \right) $$
$$- \gamma \sum_{j=h\tau}^{k-2} (\Phi^{j+1/3} - \bar{\Phi}^{j+1/3}) \prod_{i=j}^{k-1} W^i$$

$$
\left. - \gamma \sum_{j=h\tau}^{k-3} (G^{j+1/3} - \Phi^{j+1/3} - \bar{G}^{j+1/3} + \bar{\Phi}^{j+1/3}) \prod_{i=j}^{k-1} W^i \right\|_F^2 \right]
$$

$$
+ (1 + \beta_1^{-1})\gamma^2 \mathbb{E}\left[ \left\| \Phi^{(k-1)+1/3} - \bar{\Phi}^{(k-1)+1/3} \right\|_F^2 \right]
$$

$$
+ (1 + \beta_1)\gamma^2 \mathbb{E}\left[ \left\| G^{(k-2)+1/3} - \Phi^{(k-2)+1/3} - \bar{G}^{(k-2)+1/3} + \bar{\Phi}^{(k-2)+1/3} \right\|_F^2 \right]
$$

$$
+ \gamma^2 \mathbb{E}\left[ \left\| G^{(k-1)+1/3} - \Phi^{(k-1)+1/3} - \bar{G}^{(k-1)+1/3} + \bar{\Phi}^{(k-1)+1/3} \right\|_F^2 \right]
$$

$$
\leq (1+\beta_1)(1+\beta_2)\mathbb{E}\left[ \left\| Z^{h\tau} \prod_{i=h\tau}^{k-1} W^i - \bar{Z}^{h\tau} - \left( \bar{Z}^{h\tau} \prod_{i=h\tau}^{k-1} W^i - \bar{Z}^{h\tau} \right) \right.\right.
$$

$$
- \gamma \sum_{j=h\tau}^{k-3} (\Phi^{j+1/3} - \bar{\Phi}^{j+1/3}) \prod_{i=j}^{k-1} W^i
$$

$$
\left.\left. - \gamma \sum_{j=h\tau}^{k-3} (G^{j+1/3} - \Phi^{j+1/3} - \bar{G}^{j+1/3} + \bar{\Phi}^{j+1/3}) \prod_{i=j}^{k-1} W^i \right\|_F^2 \right]
$$

$$
+ (1 + \beta_1^{-1})\gamma^2 \mathbb{E}\left[ \left\| \Phi^{(k-1)+1/3} - \bar{\Phi}^{(k-1)+1/3} \right\|_F^2 \right]
$$

$$
+ (1 + \beta_1)(1 + \beta_2^{-1})\gamma^2 \mathbb{E}\left[ \left\| \Phi^{(k-2)+1/3} - \bar{\Phi}^{(k-2)+1/3} \right\|_F^2 \right]
$$

$$
+ (1 + \beta_1)\gamma^2 \mathbb{E}\left[ \left\| G^{(k-2)+1/3} - \Phi^{(k-2)+1/3} - \bar{G}^{(k-2)+1/3} + \bar{\Phi}^{(k-2)+1/3} \right\|_F^2 \right]
$$

$$
+ \gamma^2 \mathbb{E}\left[ \left\| G^{(k-1)+1/3} - \Phi^{(k-1)+1/3} - \bar{G}^{(k-1)+1/3} + \bar{\Phi}^{(k-1)+1/3} \right\|_F^2 \right].
$$

Proceeding in a similar way for all the terms, we obtain

$$
M \cdot \mathbb{E}\left[\mathrm{Err}(k)\right] \leq (1+\beta_1)\dots(1+\beta_{k-1-h\tau})\mathbb{E}\left[ \left\| Z^{h\tau} \prod_{i=h\tau}^{k-1} W^i - \bar{Z}^{h\tau} - \left( \bar{Z}^{h\tau} \prod_{i=h\tau}^{k-1} W^i - \bar{Z}^{h\tau} \right) \right\|_F^2 \right]
$$

$$
+ \gamma^2 \sum_{j=h\tau}^{k-1} (1+\beta_1)\dots(1+\beta_{k-j-1})(1+\beta_{k-j}^{-1})\mathbb{E}\left[ \left\| \Phi^{j+1/3} - \bar{\Phi}^{j+1/3} \right\|_F^2 \right]
$$

$$
+ \gamma^2 \sum_{j=h\tau}^{k-1} (1+\beta_1)\dots(1+\beta_{k-1-j})\mathbb{E}\left[ \left\| G^{j+1/3} - \Phi^{j+1/3} - \bar{G}^{j+1/3} + \bar{\Phi}^{j+1/3} \right\|_F^2 \right].
$$

$$
\tag{32}
$$

Setting $\beta_i = \frac{1}{\alpha-i}$, where $\alpha \geq 4\tau$, gives, for all $i = 0, \dots, (k-1-h\tau)$, that

$$
(1+\beta_1)(1+\beta_2)\dots(1+\beta_i) = \frac{\alpha}{\alpha-i}.
$$

By the definition of $h$, we have $k - 1 - h\tau \leq 2\tau$. Hence, for all $i = 0, \dots, (k-1-h\tau)$,

$$
(1+\beta_1)(1+\beta_2)\dots(1+\beta_i) \leq (1+\beta_1)(1+\beta_2)\dots(1+\beta_{k-1-h\tau}) \leq \frac{\alpha}{\alpha-2\tau} \leq 2.
$$

Moreover, $1 + \beta_i^{-1} \leq \alpha$. Substituting these estimates into (32), we obtain

$$
M \cdot \mathbb{E}\left[\mathrm{Err}(k)\right] \leq \frac{\alpha}{\alpha-2\tau}\mathbb{E}\left[ \left\| Z^{h\tau} \prod_{i=h\tau}^{k-1} W^i - \bar{Z}^{h\tau} - \left( \bar{Z}^{h\tau} \prod_{i=h\tau}^{k-1} W^i - \bar{Z}^{h\tau} \right) \right\|_F^2 \right]
$$

$$+ 2\gamma^2 \alpha \sum_{j=h\tau}^{k-1} \mathbb{E}\left[\left\|\Phi^{j+1/3} - \bar{\Phi}^{j+1/3}\right\|_F^2\right]$$

$$+ 2\gamma^2 \sum_{j=h\tau}^{k-1} \mathbb{E}\left[\left\|G^{j+1/3} - \Phi^{j+1/3} - \bar{G}^{j+1/3} + \bar{\Phi}^{j+1/3}\right\|_F^2\right].$$

Choosing $\alpha = 4\tau\left(1 + \frac{2}{p}\right)$, we get

$$M \cdot \mathbb{E}\left[\text{Err}(k)\right] \leq \left(1 + \frac{1}{1 + \frac{4}{p}}\right) \mathbb{E}\left[\left\|\bar{Z}^{h\tau} \prod_{i=h\tau}^{k-1} W^i - \bar{Z}^{h\tau} - \left(\bar{Z}^{h\tau} \prod_{i=h\tau}^{k-1} W^i - \bar{Z}^{h\tau}\right)\right\|_F^2\right]$$

$$+ \frac{24\gamma^2\tau}{p} \sum_{j=h\tau}^{k-1} \mathbb{E}\left[\left\|\Phi^{j+1/3} - \bar{\Phi}^{j+1/3}\right\|_F^2\right]$$

$$+ 2\gamma^2 \sum_{j=h\tau}^{k-1} \mathbb{E}\left[\left\|G^{j+1/3} - \Phi^{j+1/3} - \bar{G}^{j+1/3} + \bar{\Phi}^{j+1/3}\right\|_F^2\right].$$

Noticing that, for a matrix $A \in \mathbb{R}^{n \times M}$ with columns $A_i$, $\|A - \bar{A}\|_F^2 = \sum_{i=1}^M \|A_i - \bar{A}_i\|^2 \leq \sum_{i=1}^M \|A_i\|^2 = \|A\|_F^2$, we further obtain

$$M \cdot \mathbb{E}\left[\text{Err}(k)\right] \leq \left(1 + \frac{1}{1 + \frac{4}{p}}\right) \mathbb{E}\left[\left\|Z^{h\tau} \prod_{i=h\tau}^{k-1} W^i - \bar{Z}^{h\tau}\right\|_F^2\right]$$

$$+ \frac{24\gamma^2\tau}{p} \sum_{j=h\tau}^{k-1} \mathbb{E}\left[\left\|\Phi^{j+1/3} - \bar{\Phi}^{j+1/3}\right\|_F^2\right]$$

$$+ 2\gamma^2 \sum_{j=h\tau}^{k-1} \mathbb{E}\left[\left\|G^{j+1/3} - \Phi^{j+1/3} - \bar{G}^{j+1/3} + \bar{\Phi}^{j+1/3}\right\|_F^2\right]$$

$$\overset{\textcolor{red}{③}}{\leq} (1 - p)\left(1 + \frac{1}{1 + \frac{4}{p}}\right) \mathbb{E}\left[\left\|Z^{h\tau} - \bar{Z}^{h\tau}\right\|_F^2\right]$$

$$+ \frac{24\gamma^2\tau}{p} \sum_{j=h\tau}^{k-1} \mathbb{E}\left[\left\|\Phi^{j+1/3} - \bar{\Phi}^{j+1/3}\right\|_F^2\right]$$

$$+ 2\gamma^2 \sum_{j=h\tau}^{k-1} \mathbb{E}\left[\left\|G^{j+1/3} - \Phi^{j+1/3} - \bar{G}^{j+1/3} + \bar{\Phi}^{j+1/3}\right\|_F^2\right]. \tag{33}$$

It is easy to see that $(1 - p)\left(1 + \frac{1}{1 + \frac{4}{p}}\right) \leq (1 - p)\left(1 + \frac{p}{4}\right) \leq \left(1 - \frac{3p}{4}\right)$. It remains to estimate the last two terms in the r.h.s. of (33). For the last but one term, we have

$$\mathbb{E}\left[\left\|\Phi^{j+1/3} - \bar{\Phi}^{j+1/3}\right\|_F^2\right] = \sum_{m=1}^M \left[\mathbb{E}\left\|F_m(z_m^{j+1/3}) - \frac{1}{M}\sum_{i=1}^M F_i(z_i^{j+1/3})\right\|^2\right]$$

$$\overset{\textcolor{red}{⑪}}{\leq} 3 \sum_{m=1}^M \left[\mathbb{E}\left\|F_m(z_m^{j+1/3}) - F_m(\bar{z}^{j+1/3})\right\|^2 + \mathbb{E}\left\|F_m(\bar{z}^{j+1/3}) - \frac{1}{M}\sum_{i=1}^M F_i(\bar{z}^{j+1/3})\right\|^2\right.$$

$$\left. + \mathbb{E}\left\|\frac{1}{M}\sum_{i=1}^M F_i(\bar{z}^{j+1/3}) - \frac{1}{M}\sum_{i=1}^M F_i(z_i^{j+1/3})\right\|^2\right]$$

$$\overset{\textcolor{red}{⑤}}{\leq} 3 \sum_{m=1}^M \left[D^2 + \mathbb{E}\left\|\frac{1}{M}\sum_{i=1}^M F_i(\bar{z}^{j+1/3}) - \frac{1}{M}\sum_{i=1}^M F_i(z_i^{j+1/3})\right\|^2\right.$$

$$+ \mathbb{E}\left\|F_m(z_m^{j+1/3}) - F_m(\bar{z}^{j+1/3})\right\|^2\Bigg]$$

$$\overset{\text{(L)}}{\leq} 6ML^2\mathbb{E}\left[\text{Err}(j+1/3)\right] + 3MD^2.$$

For the last term, we have

$$\mathbb{E}\left[\left\|G^{j+1/3} - \Phi^{j+1/3} - \bar{G}^{j+1/3} + \bar{\Phi}^{j+1/3}\right\|_F^2\right]$$

$$= \sum_{m=1}^{M}\left[\mathbb{E}\left\|F_m(z_m^{j+1/3}, \xi_m^{j+1/3}) - F_m(z_m^{j+1/3}) - \frac{1}{M}\sum_{i=1}^{M}\left(F_i(z_i^{j+1/3}, \xi_i^{j+1/3}) - F_i(z_i^{j+1/3})\right)\right\|^2\right]$$

$$\overset{\text{(11)}}{\leq} 2\sum_{m=1}^{M}\left[\mathbb{E}\left\|F_m(z_m^{j+1/3}, \xi_m^{j+1/3}) - F_m(z_m^{j+1/3})\right\|^2\right.$$

$$\left.+ \mathbb{E}\left\|\frac{1}{M}\sum_{i=1}^{M}\left(F_i(z_i^{j+1/3}, \xi_i^{j+1/3}) - F_i(z_i^{j+1/3})\right)\right\|^2\right]$$

$$\overset{\text{(4)}}{\leq} 4M\sigma^2.$$

Substituting the last two bounds into (33), we obtain (30):

$$\mathbb{E}\left[\text{Err}(k)\right] \leq \left(1 - \frac{3p}{4}\right)\mathbb{E}[\text{Err}(h\tau)] + \frac{144\gamma^2 L^2 \tau}{p}\sum_{j=h\tau}^{k-1}\mathbb{E}\left[\text{Err}(j+1/3)\right] + \left(\frac{72D^2\tau}{p} + 8\sigma^2\right)\sum_{j=h\tau}^{k-1}\gamma^2.$$

The estimate for $\mathbb{E}\left[\text{Err}(k+1/3)\right]$ is obtained in a similar way. Indeed, it is sufficient to note that $M\mathbb{E}\left[\text{Err}(k+1/3)\right] = \mathbb{E}\|Z^k - \gamma G^k - \bar{Z}^k + \gamma\bar{G}^k\|_F^2$. Then, in the proof, we take $\alpha = 4\tau\left(1 + \frac{2}{p}\right) - 1$ and use also $\beta_0 = \frac{1}{\alpha}$ for the term associated with $G^k - \bar{G}^k$. In this way, we obtain $(1+\beta_0)(1+\beta_1)(1+\beta_2)\ldots(1+\beta_i) \leq (1+\beta_0)(1+\beta_1)(1+\beta_2)\ldots(1+\beta_{k-1-h\tau}) \leq \frac{\alpha+1}{\alpha-2\tau} \leq 3$, $(1+\beta_i^{-1}) \leq \alpha+1$. This gives us the final bound (31):

$$\mathbb{E}\left[\text{Err}(k+1/3)\right] \leq \left(1 - \frac{3p}{4}\right)\mathbb{E}[\text{Err}(h\tau)] + \frac{216\gamma^2 L^2 \tau}{p}\sum_{j=h\tau}^{k-1}\mathbb{E}\left[\text{Err}(j+1/3)\right] + \frac{216\gamma^2 L^2 \tau}{p}\mathbb{E}\left[\text{Err}(k)\right]$$

$$+ \left(\frac{108D^2\tau}{p} + 12\sigma^2\right)\sum_{j=h\tau}^{k-1}\gamma^2 + \left(\frac{108D^2\tau}{p} + 12\sigma^2\right)\gamma^2.$$

$\square$

We now notice that the r.h.s. of (30) and (31) involve the terms $\sum_{j=h\tau}^{k-1}\mathbb{E}\left[\text{Err}(j+1/3)\right]$. Thus, in order to resolve the recurrences in (30) and (31), we need also the bounds for $\mathbb{E}\left[\text{Err}(j+1/3)\right]$ for all $h\tau \leq j \leq k-1$, where $h = \lfloor k/\tau\rfloor - 1$. If $(h+1)\tau \leq j \leq k-1$, then we can use the same bounds (30) and (31) changing $k$ to $j$ since for such values of $j$ we have $\lfloor j/\tau\rfloor - 1 = h$. Thus, it remains to consider such $j$ that $h\tau \leq j < (h+1)\tau$. This is done in the second technical lemma of this subsection.

**Lemma C.6.** *Under Assumptions 3.1, 3.3, 3.4, 2.2 for $(h+1)\tau > j \geq h\tau$ with $h = \lfloor k/\tau\rfloor - 1$, it holds that*

$$\mathbb{E}\left[\text{Err}(j)\right] \leq \left(1 + \frac{p}{4}\right)\mathbb{E}[\text{Err}(h\tau)] + \frac{144\gamma^2 L^2 \tau}{p}\sum_{i=h\tau}^{j-1}\mathbb{E}\left[\text{Err}(i+1/3)\right]$$

$$+ \left(\frac{72D^2\tau}{p} + 8\sigma^2\right)\sum_{i=h\tau}^{j-1}\gamma^2, \tag{34}$$

$$\mathbb{E}\left[\text{Err}(j+1/3)\right] \le \left(1 + \frac{p}{4}\right)\mathbb{E}[\text{Err}(h\tau)] + \frac{216\gamma^2 L^2 \tau}{p}\sum_{i=h\tau}^{j-1}\mathbb{E}\left[\text{Err}(i+1/3)\right] + \frac{216\gamma^2 L^2 \tau}{p}\mathbb{E}\left[\text{Err}(j)\right]$$

$$+ \left(\frac{108 D^2 \tau}{p} + 12\sigma^2\right)\sum_{i=h\tau}^{j-1}\gamma^2 + \left(\frac{108 D^2 \tau}{p} + 12\sigma^2\right)\gamma^2. \tag{35}$$

**Proof:** The proof follows the same lines as the proof of Lemma C.5 until (33), which needs to be modified since in the situation of the current Lemma, we can not use (3) for small $j$'s. The modification is as follows:

$$M \cdot \mathbb{E}\left[\text{Err}(j)\right] \le \left(1 + \frac{1}{1 + \frac{4}{p}}\right)\mathbb{E}\left[\left\|\left(Z^{h\tau} - \bar{Z}^{h\tau}\right)\prod_{i=h\tau}^{j-1}W^i\right\|_F^2\right]$$

$$+ \frac{24\gamma^2\tau}{p}\sum_{i=h\tau}^{j-1}\mathbb{E}\left[\left\|\Phi^{i+1/3} - \bar{\Phi}^{i+1/3}\right\|_F^2\right]$$

$$+ 2\gamma^2\sum_{i=h\tau}^{j-1}\mathbb{E}\left[\left\|G^{i+1/3} - \Phi^{i+1/3} - \bar{G}^{i+1/3} + \bar{\Phi}^{i+1/3}\right\|_F^2\right]$$

$$\overset{(15)}{\le} \left(1 + \frac{1}{1 + \frac{4}{p}}\right)\mathbb{E}\left[\left\|Z^{h\tau} - \bar{Z}^{h\tau}\right\|_F^2\right]$$

$$+ \frac{24\gamma^2\tau}{p}\sum_{i=h\tau}^{j-1}\mathbb{E}\left[\left\|\Phi^{i+1/3} - \bar{\Phi}^{i+1/3}\right\|_F^2\right]$$

$$+ 2\gamma^2\sum_{i=h\tau}^{j-1}\mathbb{E}\left[\left\|G^{i+1/3} - \Phi^{i+1/3} - \bar{G}^{i+1/3} + \bar{\Phi}^{i+1/3}\right\|_F^2\right],$$

where we also used that $\|W^i\|_2 \le 1$. The rest of the proof is similar to the proof of Lemma C.5.

$\square$

### C.2.5 Combining the Building Blocks for the Final Bound

We are finally ready to combine the building blocks and obtain the convergence rate result for Algorithm 1 in the strongly-monotone case. We combine the refined general bound for the per-iteration progress (29) with the bounds for the consensus error terms (30), (31), (34) and (35). We note that, in general, $\mathbb{E}\left[\text{Err}(k+1/3)\right]$ may be smaller than $\mathbb{E}\left[\text{Err}(k)\right]$, but since the r.h.s. of (31) upper bounds the r.h.s. of (30), we assume, for simplicity, that $\mathbb{E}\left[\text{Err}(k+1/3)\right] \ge \mathbb{E}\left[\text{Err}(k)\right]$. We additionally use that, by the Theorem assumptions, $\gamma \le \frac{p}{120\tau L}$ and $\gamma \le \frac{1}{3L}$, and write the resulting recurrences as follows.

• Using that $\gamma \le \frac{1}{3L}$ and the assumption that $\mathbb{E}\left[\text{Err}(k+1/3)\right] \ge \mathbb{E}\left[\text{Err}(k)\right]$, the recurrence (29) transforms into

$$\mathbb{E}\left[\|\bar{z}^{k+1} - z^*\|^2\right] \le \left(1 - \frac{\gamma\mu}{2}\right)\mathbb{E}\left[\|\bar{z}^k - z^*\|^2\right]$$

$$+ \left(\frac{\gamma L^2}{\mu} + 5\gamma^2 L^2\right)\mathbb{E}\left[\text{Err}(k+1/3)\right] + 5\gamma^2 L^2 \mathbb{E}\left[\text{Err}(k)\right] + \frac{10\gamma^2\sigma^2}{M}$$

$$\le \left(1 - \frac{\gamma\mu}{2}\right)\mathbb{E}\left[\|\bar{z}^k - z^*\|^2\right] + \left(\frac{\gamma L^2}{\mu} + 10\gamma^2 L^2\right)\mathbb{E}\left[\text{Err}(k+1/3)\right] + \frac{10\gamma^2\sigma^2}{M}$$

$$\le \left(1 - \frac{\gamma\mu}{2}\right)\mathbb{E}\left[\|\bar{z}^k - z^*\|^2\right] + \frac{2\gamma L^2}{\mu}\mathbb{E}\left[\text{Err}(k+1/3)\right] + \frac{10\gamma^2\sigma^2}{M}.$$

• Using the assumption that $\mathbb{E}\left[\text{Err}(k+1/3)\right] \ge \mathbb{E}\left[\text{Err}(k)\right]$, the recurrence (31) takes the form

$$\mathbb{E}\left[\text{Err}(k+1/3)\right] \le \left(1 - \frac{3p}{4}\right)\mathbb{E}[\text{Err}(h\tau)] + \frac{216\gamma^2 L^2 \tau}{p}\sum_{j=h\tau}^{k-1}\mathbb{E}\left[\text{Err}(j+1/3)\right] + \frac{216\gamma^2 L^2 \tau}{p}\mathbb{E}\left[\text{Err}(k)\right]$$

$$+ \left( \frac{108D^2\tau}{p} + 12\sigma^2 \right) \sum_{j=h\tau}^{k-1} \gamma^2 + \left( \frac{108D^2\tau}{p} + 12\sigma^2 \right) \gamma^2$$

$$\leq \left( 1 - \frac{3p}{4} \right) \mathbb{E}[\mathrm{Err}(h\tau + 1/3)] + \frac{216\gamma^2 L^2\tau}{p} \sum_{j=h\tau}^{k-1} \mathbb{E}\left[ \mathrm{Err}(j + 1/3) \right] + \frac{216\gamma^2 L^2\tau}{p} \mathbb{E}\left[ \mathrm{Err}(k + 1/3) \right]$$

$$+ \left( \frac{108D^2\tau}{p} + 12\sigma^2 \right) \sum_{j=h\tau}^{k-1} \gamma^2 + \left( \frac{108D^2\tau}{p} + 12\sigma^2 \right) \gamma^2.$$

Rearranging, we obtain

$$\left( 1 - \frac{216\gamma^2 L^2\tau}{p} \right) \mathbb{E}\left[ \mathrm{Err}(k + 1/3) \right] \leq \left( 1 - \frac{3p}{4} \right) \mathbb{E}[\mathrm{Err}(h\tau + 1/3)] + \frac{216\gamma^2 L^2\tau}{p} \sum_{j=h\tau}^{k-1} \mathbb{E}\left[ \mathrm{Err}(j + 1/3) \right]$$

$$+ \left( \frac{216D^2\tau}{p} + 24\sigma^2 \right) \sum_{j=h\tau}^{k-1} \gamma^2.$$

Using that $\gamma \leq \frac{p}{120\tau L}$, we get

$$\left( 1 - \frac{p}{64} \right) \mathbb{E}\left[ \mathrm{Err}(k + 1/3) \right] \leq \left( 1 - \frac{3p}{4} \right) \mathbb{E}[\mathrm{Err}(h\tau + 1/3)] + \frac{p}{66\tau} \sum_{j=h\tau}^{k-1} \mathbb{E}\left[ \mathrm{Err}(j + 1/3) \right]$$

$$+ \left( \frac{216D^2\tau}{p} + 24\sigma^2 \right) \sum_{j=h\tau}^{k-1} \gamma^2.$$

Finally, using that $0 < p \leq 1$, and, hence, $\left( 1 - \frac{3p}{4} \right) \left( 1 - \frac{p}{64} \right)^{-1} \leq 1 - \frac{p}{2}$ and $\left( 1 - \frac{p}{64} \right)^{-1} \leq \frac{64}{63} \leq \frac{66}{64}$, we obtain

$$\mathbb{E}\left[ \mathrm{Err}(k + 1/3) \right] \leq \left( 1 - \frac{p}{2} \right) \mathbb{E}[\mathrm{Err}(h\tau + 1/3)] + \frac{p}{64\tau} \sum_{j=h\tau}^{k-1} \mathbb{E}\left[ \mathrm{Err}(j + 1/3) \right]$$

$$+ \left( \frac{225D^2\tau}{p} + 25\sigma^2 \right) \sum_{j=h\tau}^{k-1} \gamma^2.$$

Note that the above inequality holds for $h = \lfloor k/\tau \rfloor - 1$ since we started with (31). In the same way, for $k - 1 \geq j \geq (h+1)\tau$, since in this case $h = \lfloor j/\tau \rfloor - 1$, we have the inequality

$$\mathbb{E}\left[ \mathrm{Err}(j + 1/3) \right] \leq \left( 1 - \frac{p}{2} \right) \mathbb{E}[\mathrm{Err}(h\tau + 1/3)] + \frac{p}{64\tau} \sum_{i=h\tau}^{j-1} \mathbb{E}\left[ \mathrm{Err}(i + 1/3) \right]$$

$$+ \left( \frac{225D^2\tau}{p} + 25\sigma^2 \right) \sum_{i=h\tau}^{j-1} \gamma^2.$$

• We repeat the same derivations for (35). First,

$$\mathbb{E}\left[ \mathrm{Err}(j + 1/3) \right] \leq \left( 1 + \frac{p}{4} \right) \mathbb{E}[\mathrm{Err}(h\tau)] + \frac{216\gamma^2 L^2\tau}{p} \sum_{i=h\tau}^{j-1} \mathbb{E}\left[ \mathrm{Err}(i + 1/3) \right] + \frac{216\gamma^2 L^2\tau}{p} \mathbb{E}\left[ \mathrm{Err}(j) \right]$$

$$+ \left( \frac{108D^2\tau}{p} + 12\sigma^2 \right) \sum_{i=h\tau}^{j-1} \gamma^2 + \left( \frac{108D^2\tau}{p} + 12\sigma^2 \right) \gamma^2$$

$$\leq \left( 1 + \frac{p}{4} \right) \mathbb{E}[\mathrm{Err}(h\tau + 1/3)] + \frac{216\gamma^2 L^2\tau}{p} \sum_{i=h\tau}^{j-1} \mathbb{E}\left[ \mathrm{Err}(i + 1/3) \right] + \frac{216\gamma^2 L^2\tau}{p} \mathbb{E}\left[ \mathrm{Err}(j + 1/3) \right]$$

$$+ \left( \frac{108D^2\tau}{p} + 12\sigma^2 \right) \sum_{i=h\tau}^{j-1} \gamma^2 + \left( \frac{108D^2\tau}{p} + 12\sigma^2 \right) \gamma^2.$$

Further, rearranging, we obtain

$$\left(1 - \frac{216\gamma^2 L^2 \tau}{p}\right) \mathbb{E}\left[\text{Err}(j + 1/3)\right] \leq \left(1 + \frac{p}{4}\right) \mathbb{E}[\text{Err}(h\tau + 1/3)] + \frac{216\gamma^2 L^2 \tau}{p} \sum_{i=h\tau}^{j-1} \mathbb{E}\left[\text{Err}(i + 1/3)\right]$$

$$+ \left(\frac{216D^2\tau}{p} + 24\sigma^2\right) \sum_{i=h\tau}^{j-1} \gamma^2.$$

Since $\gamma \leq \frac{p}{120\tau L}$, we get

$$\left(1 - \frac{p}{64}\right) \mathbb{E}\left[\text{Err}(j + 1/3)\right] \leq \left(1 + \frac{p}{4}\right) \mathbb{E}[\text{Err}(h\tau + 1/3)] + \frac{p}{66\tau} \sum_{i=h\tau}^{j-1} \mathbb{E}\left[\text{Err}(i + 1/3)\right]$$

$$+ \left(\frac{216D^2\tau}{p} + 24\sigma^2\right) \sum_{i=h\tau}^{j-1} \gamma^2.$$

Since $0 < p \leq 1$, and, hence, $\left(1 + \frac{p}{4}\right)\left(1 - \frac{p}{64}\right)^{-1} \leq 1 + \frac{p}{2}$ and $\left(1 - \frac{p}{64}\right)^{-1} \leq \frac{64}{63} \leq \frac{66}{64}$, we obtain

$$\mathbb{E}\left[\text{Err}(j + 1/3)\right] \leq \left(1 + \frac{p}{2}\right) \mathbb{E}[\text{Err}(h\tau + 1/3)] + \frac{p}{64\tau} \sum_{i=h\tau}^{j-1} \mathbb{E}\left[\text{Err}(i + 1/3)\right]$$

$$+ \left(\frac{225D^2\tau}{p} + 25\sigma^2\right) \sum_{i=h\tau}^{j-1} \gamma^2.$$

Summarizing the above three bullet points, we have the following recurrences

$$\mathbb{E}\left[\|\bar{z}^{k+1} - z^*\|^2\right] \leq \left(1 - \frac{\gamma\mu}{2}\right) \mathbb{E}\left[\|\bar{z}^k - z^*\|^2\right] + \frac{2\gamma L^2}{\mu} \mathbb{E}\left[\text{Err}(k + 1/3)\right] + \frac{10\gamma^2\sigma^2}{M},$$

$$\mathbb{E}\left[\text{Err}(k + 1/3)\right] \leq \left(1 - \frac{p}{2}\right) \mathbb{E}[\text{Err}(h\tau + 1/3)] + \frac{p}{64\tau} \sum_{j=h\tau}^{k-1} \mathbb{E}\left[\text{Err}(j + 1/3)\right]$$

$$+ \left(\frac{225D^2\tau}{p} + 25\sigma^2\right) \sum_{j=h\tau}^{k-1} \gamma^2, \quad h = \lfloor k/\tau \rfloor - 1,$$

$$\mathbb{E}\left[\text{Err}(j + 1/3)\right] \leq \left(1 - \frac{p}{2}\right) \mathbb{E}[\text{Err}(h\tau + 1/3)] + \frac{p}{64\tau} \sum_{i=h\tau}^{j-1} \mathbb{E}\left[\text{Err}(i + 1/3)\right]$$

$$+ \left(\frac{225D^2\tau}{p} + 25\sigma^2\right) \sum_{i=h\tau}^{j-1} \gamma^2, \quad k - 1 \geq j \geq (h + 1)\tau,$$

$$\mathbb{E}\left[\text{Err}(j + 1/3)\right] \leq \left(1 + \frac{p}{2}\right) \mathbb{E}[\text{Err}(h\tau + 1/3)] + \frac{p}{64\tau} \sum_{i=h\tau}^{j-1} \mathbb{E}\left[\text{Err}(i + 1/3)\right]$$

$$+ \left(\frac{225D^2\tau}{p} + 25\sigma^2\right) \sum_{i=h\tau}^{j-1} \gamma^2, \quad h\tau \leq j < (h + 1)\tau.$$

To simplify the further derivations, we introduce shortcut notations: $r_k = \mathbb{E}\left[\|\bar{z}^k - z^*\|^2\right]$, $e_k = \mathbb{E}\left[\text{Err}(k + 1/3)\right]$, $a = \frac{\mu}{2}$, $A = \frac{225D^2\tau}{p} + 25\sigma^2$, $B = \frac{2L^2}{\mu}$, $C = \frac{10\sigma^2}{M}$, and $h = \lfloor k/\tau \rfloor - 1$. Then, the previous recurrences can be written as

$$r_{k+1} \leq (1 - \gamma a) r_k + \gamma B e_k + \gamma^2 C, \tag{36}$$

$$e_k \leq \left(1 - \frac{p}{2}\right) e_{h\tau} + \frac{p}{64\tau} \sum_{j=h\tau}^{k-1} e_j + A \sum_{j=h\tau}^{k-1} \gamma^2, \quad h = \lfloor k/\tau \rfloor - 1, \tag{37}$$

$$e_j \leq \left(1 - \frac{p}{2}\right) e_{h\tau} + \frac{p}{64\tau} \sum_{i=h\tau}^{j-1} e_i + A \sum_{i=h\tau}^{j-1} \gamma^2, \quad k-1 \geq j \geq (h+1)\tau, \tag{38}$$

$$e_j \leq \left(1 + \frac{p}{2}\right) e_{h\tau} + \frac{p}{64\tau} \sum_{i=h\tau}^{j-1} e_i + A \sum_{i=h\tau}^{j-1} \gamma^2, \quad h\tau \leq j < (h+1)\tau. \tag{39}$$

The last two recurrent inequalities can be resolved w.r.t. $e_k$ using the following lemma.

**Lemma C.7.** *If a non-negative sequence $\{e_k\}_{k\geq 0}$ satisfies* (37), (38), *and* (39) *with some constants $0 < p \leq 1$, $\tau \geq 1$, $A \geq 0$, then it holds that*

$$e_k \leq \frac{16\gamma^2 A\tau}{p}.$$

**Proof:** We depart from (37) and iteratively substitute all $e_j$ for $j \geq (h+1)\tau$ starting from $k-1$ and finishing with $(h+1)\tau$:

$$e_k \leq \left(1 - \frac{p}{2}\right) \cdot \left(1 + \frac{p}{64\tau}\right) e_{h\tau} + \frac{p}{64\tau}\left(1 + \frac{p}{64\tau}\right) \sum_{j=h\tau}^{k-2} e_j + A\sum_{j=h\tau}^{k-1} \gamma^2 + \frac{p}{64\tau} \cdot A \sum_{j=h\tau}^{k-2} \gamma^2$$

$$\leq \left(1 - \frac{p}{2}\right) \cdot \left(1 + \frac{p}{64\tau}\right)^\tau e_{h\tau} + \frac{p}{64\tau}\left(1 + \frac{p}{64\tau}\right)^\tau \sum_{j=h\tau}^{(h+1)\tau-1} e_j$$

$$+ A\left(1 + \frac{p}{64\tau}\right)^{k-(h+1)\tau} \sum_{j=h\tau}^{(h+1)\tau-1} \gamma^2 + A \sum_{j=(h+1)\tau}^{k-1}\left(1 + \frac{p}{64\tau}\right)^{k-1-j}\gamma^2.$$

Next, using (39), we substitute all $e_j$ for $j$ such that $h\tau \leq j < (h+1)\tau$:

$$e_k \leq \left(1 - \frac{p}{2} + \frac{p}{64\tau}\left(1 + \frac{p}{2}\right)\right) \cdot \left(1 + \frac{p}{64\tau}\right)^\tau e_{h\tau} + \frac{p}{64\tau}\left(1 + \frac{p}{64\tau}\right)^{\tau+1} \sum_{j=h\tau}^{(h+1)\tau-2} e_j$$

$$+ A\left(1 + \frac{p}{64\tau}\right)^{k-(h+1)\tau+1} \sum_{j=h\tau}^{(h+1)\tau-2} \gamma^2 + A \sum_{j=(h+1)\tau-1}^{k-1}\left(1 + \frac{p}{64\tau}\right)^{k-1-j}\gamma^2.$$

Since $\frac{p}{64\tau}\left(1 + \frac{p}{2}\right) \leq \frac{p}{16\tau}\left(1 - \frac{p}{2}\right)$, we get

$$e_k \leq \left(1 - \frac{p}{2}\right)\left(1 + \frac{p}{16\tau}\right)\left(1 + \frac{p}{64\tau}\right)^\tau e_{h\tau} + \frac{p}{64\tau}\left(1 + \frac{p}{64\tau}\right)^{\tau+1} \sum_{j=h\tau}^{(h+1)\tau-2} e_j$$

$$+ A\left(1 + \frac{p}{64\tau}\right)^{k-(h+1)\tau+1} \sum_{j=h\tau}^{(h+1)\tau-2} \gamma^2 + A \sum_{j=(h+1)\tau-1}^{k-1}\left(1 + \frac{p}{64\tau}\right)^{k-1-j}\gamma^2.$$

Proceeding in the same way for the rest of $e_j$, we have

$$e_k \leq \left(1 - \frac{p}{2}\right)\left(1 + \frac{p}{16\tau}\right)^{2\tau} e_{h\tau} + A \sum_{j=h\tau}^{k-1}\left(1 + \frac{p}{64\tau}\right)^{k-1-j}\gamma^2.$$

Noting that $\left(1 + \frac{p}{64\tau}\right)^{k-1-j} \leq \left(1 + \frac{p}{16\tau}\right)^{2\tau} \leq \exp(p/8) \leq 1 + \frac{p}{4}$ for $p \leq 1$, we further derive

$$e_k \leq \left(1 - \frac{p}{4}\right) e_{h\tau} + 2A \sum_{j=h\tau}^{k-1} \gamma^2 \leq \left(1 - \frac{p}{4}\right) e_{h\tau} + 2A\gamma^2(k - h\tau) \leq \left(1 - \frac{p}{4}\right) e_{h\tau} + 4A\gamma^2\tau,$$

using also that $h = \lfloor k/\tau \rfloor - 1$. It remains to apply the recursion for $e_{h\tau}$ (with $e_0 = 0$), and obtain

$$e_k \le \left(1 - \frac{p}{4}\right)^2 e_{(h-1)\tau} + \left(1 - \frac{p}{4}\right) \cdot 4A\gamma^2\tau + 4A\gamma^2\tau \le \dots$$

$$\le 4A\gamma^2\tau \sum_{j=0}^{h} \left(1 - \frac{p}{4}\right)^j \le 4A\gamma^2\tau \sum_{j=0}^{\infty} \left(1 - \frac{p}{4}\right)^j$$

$$\le \frac{16\gamma^2 A\tau}{p}.$$

This finishes the proof of the bound for $e_k$.

$\square$

We proceed with the proof of the main estimate for the strongly-monotone case by substituting the above estimate for $e_k$ into (36):

$$r_{k+1} \le (1 - \gamma a)\, r_k + \frac{16\gamma^3 AB\tau}{p} + \gamma^2 C.$$

Running the recursion from 0 to $K$ gives

$$r_{K+1} \le (1 - \gamma a)^K r_0 + \frac{16\gamma^2 AB\tau}{ap} + \frac{\gamma C}{a} \le \exp\left(-\gamma a K\right) r_0 + \frac{16\gamma^2 AB\tau}{ap} + \frac{\gamma C}{a}. \qquad (40)$$

Throughout the proof of this bound we used that $\gamma \le \frac{p}{120\tau L}$ and $\gamma \le \frac{1}{3L}$. Let us denote $\frac{1}{d} = \frac{p}{120L\tau}$. By the definitions of $\tau$ and $p$, we have that $\frac{1}{d} = \frac{p}{120L\tau} < \frac{1}{3L}$, and the choice

$$\gamma = \min\left\{\frac{p}{120\tau L}, \frac{2\ln\left(\max\{2, \mu^2 M r_0 K/(40\sigma^2)\}\right)}{\mu K}\right\} = \min\left\{\frac{1}{d}, \frac{\ln\left(\max\{2, a^2 r_0 K/C\}\right)}{aK}\right\}$$

in the Theorem assumptions is valid to obtain (40). Further, this choice leads to the following convergence rate estimates.

- If $\frac{1}{d} \ge \frac{\ln\left(\max\{2, a^2 r_0 K/C\}\right)}{aK}$ then $\gamma = \frac{\ln\left(\max\{2, a^2 r_0 K/C\}\right)}{aK}$ gives

$$\mathcal{O}\left(\exp\left(-a^2 K \cdot \frac{\ln\left(\max\{2, a^2 r_0 K/C\}\right)}{aK}\right) r_0 + \frac{AB\tau}{ap} \cdot \frac{\ln^2\left(\max\{2, a^2 r_0 K/C\}\right)}{a^2 K^2} + \frac{C}{a} \cdot \frac{\ln\left(\max\{2, a^2 r_0 K/C\}\right)}{aK}\right)$$

$$= \tilde{\mathcal{O}}\left(\exp\left(-\ln\left(\max\{2, a^2 r_0 K/C\}\right)\right) r_0 + \frac{AB\tau}{a^3 pK^2} + \frac{C}{a^2 K}\right) \le \tilde{\mathcal{O}}\left(\frac{C}{a^2 K} + \frac{AB\tau}{a^3 pK^2} + \frac{C}{a^2 K}\right).$$

- If $\frac{1}{d} \le \frac{\ln\left(\max\{2, a^2 r_0 K/C\}\right)}{aK}$ then $\gamma = \frac{1}{d}$ gives

$$\mathcal{O}\left(\exp\left(-aK \cdot \frac{1}{d}\right) r_0 + \frac{AB\tau}{ap} \cdot \frac{1}{d^2} + \frac{C}{a} \cdot \frac{1}{d}\right)$$

$$\le \mathcal{O}\left(\exp\left(-\frac{aK}{d}\right) r_0 + \frac{AB\tau}{ap} \cdot \frac{\ln^2\left(\max\{2, a^2 r_0 K/C\}\right)}{a^2 K^2} + \frac{C}{a} \cdot \frac{\ln\left(\max\{2, a^2 r_0 K/C\}\right)}{aK}\right)$$

$$= \tilde{\mathcal{O}}\left(\exp\left(-\frac{aK}{d}\right) r_0 + \frac{AB\tau}{a^3 pK^2} + \frac{C}{a^2 K}\right).$$

Thus, our choice of $\gamma$ leads to the desired estimates for the convergence rate

$$r_{k+1} = \tilde{\mathcal{O}}\left(\exp\left(-\frac{aK}{d}\right) r_0 + \frac{AB\tau}{a^3 pK^2} + \frac{C}{a^2 K}\right).$$

Finally, we substitute constants $A, B, C, a, d$ and obtain

$$r_{k+1} = \tilde{\mathcal{O}}\left(\exp\left(-\frac{p\mu K}{240\tau L}\right) r_0 + \frac{\tau^2 D^2 L^2}{p^2 \mu^4 K^2} + \frac{\tau\sigma^2 L^2}{p\mu^4 K^2} + \frac{\sigma^2}{\mu^2 MK}\right).$$

Using the definition $\Delta = \frac{\tau}{p}\left(\frac{D^2\tau}{p} + \sigma^2\right)$, from the last estimate we obtain the bound in the theorem statement.

This completes the proof in the strongly-monotone case.

$\square$

## C.3 Proof of Theorem 4.1, Monotone Case

The proof partially relies on the general per-iterate estimate in the previous section, which we slightly modify and refine for the purposes of this section. We first note that in the proof of inequality (24) we can take an arbitrary $z$ instead of $z^*$. Rearranging the terms, we obtain for an arbitrary $z$:

$$2\gamma\mathbb{E}\left[\langle \bar{g}^{k+1/3}, \bar{z}^{k+1/3} - z\rangle\right] \leq \mathbb{E}\left[\|\bar{z}^k - z\|^2\right] - \mathbb{E}\left[\|\bar{z}^{k+1} - z\|^2\right]$$
$$-\mathbb{E}\left[\|\bar{z}^{k+1/3} - \bar{z}^k\|^2\right] + \gamma^2\mathbb{E}\left[\|\bar{g}^{k+1/3} - \bar{g}^k\|^2\right]. \quad (41)$$

Next, we need two bounds: a lower bound for the l.h.s. that relates it with the true operator $F$, and an upper bound for the last term in the r.h.s. that is given by Lemma C.4.

The lower bound is given by the following result.

**Lemma C.8.** *Let the operator $F$ satisfy Assumptions 3.1, 3.3. Then, for any fixed $z$, we have*

$$\mathbb{E}\left[\langle \bar{g}^{k+1/3}, \bar{z}^{k+1/3} - z\rangle\right] \geq \mathbb{E}\left[\langle F(\bar{z}^{k+1/3}), \bar{z}^{k+1/3} - z\rangle\right] \quad (42)$$

$$-\frac{\gamma L^2}{2}\mathbb{E}\left[\|\bar{z}^{k+1/3} - \bar{z}^k\|^2\right] - \frac{1}{2\gamma}\mathbb{E}\mathrm{Err}(k+1/3) - L\sqrt{\mathbb{E}\mathrm{Err}(k+1/3)}\sqrt{\mathbb{E}\|\bar{z}^k - z\|^2} \quad (43)$$

**Proof:** We take into account the independence of all random vectors $\xi^i = (\xi_1^i, \ldots, \xi_m^i)$ and take only the conditional expectation $\mathbb{E}_{\xi^{k+1/3}}$ w.r.t. the vector $\xi^{k+1/3}$ conditioned on all the other randomness:

$$\mathbb{E}\left[\langle \bar{g}^{k+1/3}, \bar{z}^{k+1/3} - z\rangle\right] \overset{(17)}{=} \mathbb{E}\left[\left\langle \frac{1}{M}\sum_{m=1}^{M}\mathbb{E}_{\xi^{k+1/3}}[F_m(z_m^{k+1/3}, \xi_m^{k+1/3})], \bar{z}^{k+1/3} - z\right\rangle\right]$$

$$\overset{(4)}{=} \mathbb{E}\left[\left\langle \frac{1}{M}\sum_{m=1}^{M}F_m(z_m^{k+1/3}), \bar{z}^{k+1/3} - z\right\rangle\right]$$

$$= \mathbb{E}\left[\left\langle \frac{1}{M}\sum_{m=1}^{M}F_m(\bar{z}^{k+1/3}), \bar{z}^{k+1/3} - z\right\rangle\right]$$

$$-\mathbb{E}\left[\left\langle \frac{1}{M}\sum_{m=1}^{M}[F_m(\bar{z}^{k+1/3}) - F_m(z_m^{k+1/3})], \bar{z}^{k+1/3} - z\right\rangle\right]$$

$$= \mathbb{E}\left[\left\langle F(\bar{z}^{k+1/3}), \bar{z}^{k+1/3} - z\right\rangle\right]$$

$$-\mathbb{E}\left[\left\langle \frac{1}{M}\sum_{m=1}^{M}[F_m(\bar{z}^{k+1/3}) - F_m(z_m^{k+1/3})], \bar{z}^{k+1/3} - z\right\rangle\right].$$

Next, we estimate from below the last term in the r.h.s. Using (12) with $c = \gamma L^2$, we obtain

$$-\mathbb{E}\left\langle \frac{1}{M}\sum_{m=1}^{M}[F_m(\bar{z}^{k+1/3}) - F_m(z_m^{k+1/3})], \bar{z}^{k+1/3} - \bar{z}^k + \bar{z}^k - z\right\rangle$$

$$\geq -\frac{\gamma L^2}{2}\mathbb{E}\|\bar{z}^{k+1/3} - \bar{z}^k\|^2 - \frac{1}{2\gamma L^2}\mathbb{E}\left\|\frac{1}{M}\sum_{m=1}^{M}[F_m(\bar{z}^{k+1/3}) - F_m(z_m^{k+1/3})]\right\|^2$$

$$-\mathbb{E}\left[\left\|\frac{1}{M}\sum_{m=1}^{M}[F_m(\bar{z}^{k+1/3}) - F_m(z_m^{k+1/3})]\right\|\|\bar{z}^k - z\|\right]$$

$$\overset{(L)}{\geq} -\frac{\gamma L^2}{2}\mathbb{E}\left[\|\bar{z}^{k+1/3} - \bar{z}^k\|^2\right] - \frac{L^2}{2M\gamma L^2}\mathbb{E}\left[\sum_{m=1}^{M}\left\|\bar{z}^{k+1/3} - z_m^{k+1/3}\right\|^2\right]$$

$$-\frac{L}{M}\mathbb{E}\left[\sum_{m=1}^{M}\left\|\bar{z}^{k+1/3} - z_m^{k+1/3}\right\|\|\bar{z}^k - z\|\right]$$

$$\overset{(23)}{\geq} \quad -\frac{\gamma L^2}{2}\mathbb{E}\left[\|\bar{z}^{k+1/3} - \bar{z}^k\|^2\right] - \frac{1}{2\gamma}\mathbb{E}\mathrm{Err}(k+1/3) - L\sqrt{\mathbb{E}\mathrm{Err}(k+1/3)}\sqrt{\mathbb{E}\|\bar{z}^k - z\|^2},$$

where in the last inequality we used also that

$$\mathbb{E}\left[\frac{1}{M}\sum_{m=1}^{M}\left\|\bar{z}^{k+1/3} - z_m^{k+1/3}\right\|\|\bar{z}^k - z\|\right] \leq \sqrt{\mathbb{E}\left(\frac{1}{M}\sum_{m=1}^{M}\left\|\bar{z}^{k+1/3} - z_m^{k+1/3}\right\|\right)^2}\sqrt{\mathbb{E}\|\bar{z}^k - z\|^2}$$

$$\leq \sqrt{\frac{1}{M}\mathbb{E}\sum_{m=1}^{M}\left\|\bar{z}^{k+1/3} - z_m^{k+1/3}\right\|^2}\sqrt{\mathbb{E}\|\bar{z}^k - z\|^2}$$

$$\overset{(23)}{=} \sqrt{\mathbb{E}\mathrm{Err}(k+1/3)}\sqrt{\mathbb{E}\|\bar{z}^k - z\|^2}.$$

Combining the above, we obtain the statement of the Lemma.

$\square$

Combining inequality (41) with Lemma C.4 and Lemma C.8, rearranging the terms, and using the monotonicity of the operator $F$, i.e. (SM) with $\mu = 0$, we obtain, for any $z$

$$2\gamma\mathbb{E}\left[\left\langle F(z), \bar{z}^{k+1/3} - z\right\rangle\right] \leq 2\gamma\mathbb{E}\left[\left\langle F(\bar{z}^{k+1/3}), \bar{z}^{k+1/3} - z\right\rangle\right]$$

$$\leq \mathbb{E}\left[\|\bar{z}^k - z\|^2\right] - \mathbb{E}\left[\|\bar{z}^{k+1} - z\|^2\right]$$

$$-\mathbb{E}\left[\|\bar{z}^{k+1/3} - \bar{z}^k\|^2\right] + 5\gamma^2 L^2\mathbb{E}\left[\|\bar{z}^{k+1/3} - \bar{z}^k\|^2\right]$$

$$+\frac{10\sigma^2\gamma^2}{M} + 5L^2\gamma^2\mathbb{E}\left[\mathrm{Err}(k+1/3)\right] + 5L^2\gamma^2\mathbb{E}\left[\mathrm{Err}(k)\right]$$

$$+L^2\gamma^2\mathbb{E}\left[\|\bar{z}^{k+1/3} - \bar{z}^k\|^2\right] + \mathrm{Err}(k+1/3)$$

$$+2\gamma L\sqrt{\mathbb{E}\mathrm{Err}(k+1/3)}\sqrt{\mathbb{E}\|\bar{z}^k - z\|^2}$$

$$\leq \mathbb{E}\left[\|\bar{z}^k - z\|^2\right] - \mathbb{E}\left[\|\bar{z}^{k+1} - z\|^2\right] + \frac{10\sigma^2\gamma^2}{M}$$

$$+5L^2\gamma^2\mathbb{E}\left[\mathrm{Err}(k+1/3)\right] + 5L^2\gamma^2\mathbb{E}\left[\mathrm{Err}(k)\right]$$

$$+\mathrm{Err}(k+1/3) + 2\gamma L\sqrt{\mathbb{E}\mathrm{Err}(k+1/3)}\sqrt{\mathbb{E}\|\bar{z}^k - z\|^2}$$

$$\leq \mathbb{E}\left[\|\bar{z}^k - z\|^2\right] - \mathbb{E}\left[\|\bar{z}^{k+1} - z\|^2\right] + \frac{10\sigma^2\gamma^2}{M} + (1 + 5\gamma^2 L^2)\mathbb{E}\left[\mathrm{Err}(k+1/3)\right]$$

$$+5\gamma^2 L^2\mathbb{E}\left[\mathrm{Err}(k)\right] + 2\gamma L\sqrt{\mathbb{E}\mathrm{Err}(k+1/3)}\sqrt{\mathbb{E}\|\bar{z}^k - z\|^2}, \tag{44}$$

where in the last but one inequality we used that, by the Theorem assumptions, $\gamma \leq \frac{1}{L\sqrt{6}}$. The above is a refined general bound for the per-iteration progress in the monotone setting. Further, by Lemma C.7, we have, for any $z$,

$$2\gamma\mathbb{E}\left[\left\langle F(z), \bar{z}^{k+1/3} - z\right\rangle\right] \leq \mathbb{E}\left[\|\bar{z}^k - z\|^2\right] - \mathbb{E}\left[\|\bar{z}^{k+1} - z\|^2\right]$$

$$+\frac{10\sigma^2\gamma^2}{M} + (1 + 5\gamma^2 L^2)\cdot\frac{8\gamma^2\tau}{p}\cdot\left(\frac{225D^2\tau}{p} + 25\sigma^2\right)$$

$$+5\gamma^2 L^2\cdot\frac{8\gamma^2\tau}{p}\cdot\left(\frac{225D^2\tau}{p} + 25\sigma^2\right)$$

$$+2\gamma L\sqrt{\frac{8\gamma^2\tau}{p}\cdot\left(\frac{225D^2\tau}{p} + 25\sigma^2\right)}\sqrt{\mathbb{E}\|\bar{z}^k - z\|^2}$$

$$\leq \mathbb{E}\left[\|\bar{z}^k - z\|^2\right] - \mathbb{E}\left[\|\bar{z}^{k+1} - z\|^2\right] + \frac{10\sigma^2\gamma^2}{M}$$

$$+(1 + 10\gamma^2 L^2)\cdot\frac{8\gamma^2\tau}{p}\cdot\left(\frac{225D^2\tau}{p} + 25\sigma^2\right)$$

$$+ \gamma L \sqrt{\frac{32\gamma^2\tau}{p} \cdot \left(\frac{225D^2\tau}{p} + 25\sigma^2\right)} \sqrt{\mathbb{E}\|\bar{z}^k - z\|^2}$$

$$\leq \quad \mathbb{E}\left[\|\bar{z}^k - z\|^2\right] - \mathbb{E}\left[\|\bar{z}^{k+1} - z\|^2\right] + \xi$$

$$+\sqrt{\eta}\sqrt{\mathbb{E}\|\bar{z}^k - z\|^2}, \tag{45}$$

where we denote $\Delta := 32 \cdot \frac{\tau}{p} \cdot \left(\frac{225D^2\tau}{p} + 25\sigma^2\right)$, $\xi := (1 + 10\gamma^2 L^2)\gamma^2\Delta + \frac{10\sigma^2\gamma^2}{M}$, $\eta = \gamma^4 L^2\Delta$. Our next goal is to analyze the recurrence (45) in two situations: bounded iterates and unbounded iterates.

### C.3.1  Unbounded Iterates

First, we consider the general case when the iterates $\bar{z}^k$ are not assumed to be bounded. We carefully analyze this sequence and prove that this sequence can not go too far from any solution to the variational inequality. This allows us to obtain the final convergence rate bound. Let us denote $r_k(z) := \sqrt{\mathbb{E}\|\bar{z}^k - z\|^2}$ and let $z^*$ be a solution to the variational inequality. Then, we have

$$r_k(z) \leq \sqrt{2\mathbb{E}\|\bar{z}^k - z^*\|^2 + 2\|z - z^*\|^2} \leq \sqrt{2\mathbb{E}\|\bar{z}^k - z^*\|^2} + \sqrt{2\|z - z^*\|^2}$$

$$= \sqrt{2}r_k(z^*) + \sqrt{2}\|z - z^*\|,$$

$$(r_k(z))^2 \leq 2\mathbb{E}\|\bar{z}^k - z^*\|^2 + \|z - z^*\|^2 = 2(r_k(z^*))^2 + 2\|z - z^*\|^2.$$

Thus, from (45), we have, for any $z$ and any $k \geq 0$,

$$2\gamma\mathbb{E}\left[\left\langle F(z), \bar{z}^{k+1/3} - z\right\rangle\right] \leq r_k(z)^2 - r_{k+1}(z)^2 + \xi + \sqrt{\eta}r_k(z). \tag{46}$$

Summing these inequalities from $k = 0$ to $K$, we obtain, for any $z$,

$$2\gamma(K+1)\mathbb{E}\left[\left\langle F(z), \hat{z}^K - z\right\rangle\right] \leq r_0(z)^2 + (K+1)\xi + \sqrt{\eta}\sum_{k=0}^{K} r_k(z)$$

$$\leq 2r_0(z^*)^2 + 2\|z - z^*\|^2 + (K+1)\xi$$

$$+\sqrt{\eta}\left(\sqrt{2}(K+1)\|z - z^*\| + \sqrt{2}\sum_{k=0}^{K} r_k(z^*)\right), \tag{47}$$

where $\hat{z}^K = \frac{1}{K+1}\sum_{k=0}^{K} \bar{z}^{k+1/3}$.

Our next goal is to bound from above the expression

$$r_0(z^*)^2 + (K+1)\xi + \sqrt{2\eta}\sum_{k=0}^{K} r_k(z^*).$$

Taking $z = z^*$ in (46) and using the fact that $z^*$ is a solution to the variational inequality, we obtain, for any $k \geq 0$

$$0 \leq 2\gamma\mathbb{E}\left[\left\langle F(z^*), \bar{z}^{k+1/3} - z^*\right\rangle\right] \leq r_k(z^*)^2 - r_{k+1}(z^*)^2 + \xi + \sqrt{\eta}r_k(z^*).$$

Thus, for all $k \geq 0$,

$$r_{k+1}(z^*)^2 \leq r_k(z^*)^2 + \xi + \sqrt{\eta}r_k(z^*).$$

Summing these inequalities from $k = 0$ to $K$, we obtain

$$r_{K+1}(z^*)^2 \leq r_0(z^*)^2 + (K+1)\xi + \sqrt{\eta}\sum_{k=0}^{K} r_k(z^*).$$

Note that this inequality holds for arbitrary $K \geq 0$. We next use the following technical result.

**Lemma C.9** (Lemma B.2 in [29]). *Let* $\alpha, a_0, \ldots, a_{N-1}, b, R_1, \ldots, R_{N-1}$ *be non-negative numbers and*

$$R_l \leqslant \sqrt{2} \cdot \sqrt{\left( \sum_{k=0}^{l-1} a_k + b\alpha \sum_{k=1}^{l-1} R_k \right)} \quad l = 1, \ldots, N.$$

*Then, for* $l = 1, \ldots, N$,

$$\sum_{k=0}^{l-1} a_k + b\alpha \sum_{k=1}^{l-1} R_k \leqslant \left( \sqrt{\sum_{k=0}^{l-1} a_k} + \sqrt{2}b\alpha l \right)^2.$$

Choosing $\alpha = 1$, $b = \sqrt{\eta}$, $a_0 = r_0(z^*)^2 + \xi$, $a_k = \xi$, $k = 1, ..., K - 1$, $R_k = r_k(z^*)$, we obtain

$$
\begin{aligned}
r_0(z^*)^2 + (K+1)\xi + \sqrt{\eta} \sum_{k=0}^{K} r_k(z^*) &\leq \left( \sqrt{r_0(z^*)^2 + (K+1)\xi} + (K+1)\sqrt{2\eta} \right)^2 \\
&\leq 2r_0(z^*)^2 + 2(K+1)\xi + 4(K+1)^2\eta.
\end{aligned}
$$

Combining the last inequality with (47), we obtain

$$
\begin{aligned}
2\gamma(K+1)\mathbb{E}\left[ \langle F(z), \widehat{z}^K - z \rangle \right] &\leq r_0(z^*)^2 + 2\|z - z^*\|^2 + \sqrt{2\eta}(K+1)\|z - z^*\| \\
&\quad + \left( 2r_0(z^*)^2 + 2(K+1)\xi + 4(K+1)^2\eta \right) \\
&\leq 3r_0(z^*)^2 + 2\|z - z^*\|^2 + 2(K+1)\xi \\
&\quad + \sqrt{2\eta}(K+1)\|z - z^*\| + 6(K+1)^2\eta.
\end{aligned}
$$

Dividing both sides of the inequality by $2\gamma(K+1)$ and using the definitions $\Delta := 32 \cdot \frac{\tau}{p} \cdot \left( \frac{225D^2\tau}{p} + 25\sigma^2 \right)$, $\xi := (1 + 10\gamma^2 L^2)\gamma^2\Delta + \frac{10\sigma^2\gamma^2}{M}$, $\eta := \gamma^4 L^2\Delta$, we obtain, for all $z \in \mathcal{C}$

$$
\begin{aligned}
\mathbb{E}\left[ \langle F(z), \widehat{z}^K - z \rangle \right] &\leq 2\frac{\|z^0 - z^*\|^2 + \|z - z^*\|^2}{\gamma(K+1)} + \frac{\xi}{\gamma} + \|z - z^*\|\sqrt{\frac{\eta}{2\gamma^2}} + 3(K+1)\frac{\eta}{\gamma} \\
&\leq 2\frac{\|z^0 - z^*\|^2 + \|z - z^*\|^2}{\gamma(K+1)} + \frac{10\sigma^2\gamma}{M} + (1 + 10\gamma^2 L^2)\gamma\Delta \\
&\quad + \gamma L\|z - z^*\|\sqrt{\Delta} + 3(K+1)\gamma^3 L^2\Delta \\
&\leq \frac{4\Omega_{\mathcal{C}}^2}{\gamma(K+1)} + \frac{10\sigma^2\gamma}{M} + \gamma\Delta \\
&\quad + \gamma L\Omega_{\mathcal{C}}\sqrt{\Delta} + 8(K+1)\gamma^3 L^2\Delta,
\end{aligned}
$$

where in the last inequality we used that $z^0, z, z^* \in \mathcal{C}$, $\max_{z, z' \in \mathcal{C}} \|z - z'\| \leq \Omega_{\mathcal{C}}$, and that $K \geq 1$. Our choice

$$\gamma = \min\left\{ \frac{1}{3L}, \left( \frac{2\Omega_{\mathcal{C}}^2 M}{5(K+1)\sigma^2} \right)^{\frac{1}{2}}, \left( \frac{\Omega_{\mathcal{C}}^2}{6(K+1)^2 L^2\Delta} \right)^{\frac{1}{4}} \right\},$$

in the Theorem assumptions implies

$$\frac{4\Omega_{\mathcal{C}}^2}{\gamma(K+1)} = \mathcal{O}\left( \frac{L\Omega_{\mathcal{C}}^2}{K} + \frac{\sigma\Omega_{\mathcal{C}}}{\sqrt{MK}} + \frac{\sqrt{L\Omega_{\mathcal{C}}^3\sqrt{\Delta}}}{\sqrt{K}} \right),$$

and we obtain

$$\sup_{z \in \mathcal{C}} \mathbb{E}\left[ \langle F(z), \widehat{z}^K - z \rangle \right] = \mathcal{O}\left( \frac{L\Omega_{\mathcal{C}}^2}{K} + \frac{\sigma\Omega_{\mathcal{C}}}{\sqrt{MK}} + \frac{\sqrt{L\Omega_{\mathcal{C}}^3\sqrt{\Delta}}}{\sqrt{K}} + \sqrt{\frac{(\Delta + L^2\Omega_{\mathcal{C}}^2)\Omega_{\mathcal{C}}\sqrt{\Delta}}{KL}} \right).$$

### C.3.2 Bounded Iterates

Let us now consider the situation under the additional assumption that for all $k$ the iterations of the algorithm satisfy $\|\bar{z}^k\| \leq \Omega$. In this case, summing (45) from $k = 0$ to $K$, we obtain, for any $z$,

$$
2\gamma(K+1)\mathbb{E}\left[\langle F(z), \widehat{z}^K - z\rangle\right] \leq \|z^0 - z\|^2 + (K+1)\xi + \sqrt{\eta}\sum_{k=0}^{K}\sqrt{\mathbb{E}\|\bar{z}^k - z\|^2}
$$
$$
\leq \|z^0 - z\|^2 + (K+1)\xi + 2(K+1)\sqrt{\eta}(\Omega + \|z\|).
$$

Dividing both sides of this inequality by $2\gamma(K+1)$ and using the definitions $\Delta := 32 \cdot \frac{\tau}{p} \cdot \left(\frac{225D^2\tau}{p} + 25\sigma^2\right)$, $\xi := (1 + 10\gamma^2 L^2)\gamma^2\Delta + \frac{10\sigma^2\gamma^2}{M}$, $\eta := \gamma^4 L^2\Delta$, we obtain, for all $z \in \mathcal{C}$

$$
\mathbb{E}\left[\langle F(z), \widehat{z}^K - z\rangle\right] \leq \frac{\|z^0 - z\|^2}{2\gamma(K+1)} + \frac{\xi}{\gamma} + (\Omega + \|z\|)\sqrt{\frac{\eta}{\gamma^2}}
$$
$$
\leq \frac{\|z^0 - z\|^2}{2\gamma(K+1)} + \frac{10\sigma^2\gamma}{M} + (1 + 10\gamma^2 L^2)\gamma\Delta
$$
$$
+ (\Omega + \|z\|)\gamma L\sqrt{\Delta}
$$
$$
\leq \frac{\Omega_{\mathcal{C}}^2}{2\gamma(K+1)} + \frac{10\sigma^2\gamma}{M} + 10\gamma^3 L^2\Delta
$$
$$
+ \gamma((\Omega + \Omega_{\mathcal{C}})L\sqrt{\Delta} + \Delta),
$$

where in the last inequality we used that $z^0, z, z^* \in \mathcal{C}$, $\max_{z,z' \in \mathcal{C}}\|z - z'\| \leq \Omega_{\mathcal{C}}$, and that $K \geq 1$.

Our choice

$$
\gamma = \min\left\{\frac{1}{3L}, \left(\frac{\Omega_{\mathcal{C}}^2 M}{20(K+1)\sigma^2}\right)^{\frac{1}{2}}, \left(\frac{\Omega_{\mathcal{C}}^2}{60(K+1)^2 L^2\Delta}\right)^{\frac{1}{4}}, \left(\frac{\Omega_{\mathcal{C}}^2}{(K+1)((\Omega + \Omega_{\mathcal{C}})L\sqrt{\Delta} + \Delta)}\right)^{\frac{1}{2}}\right\},
$$

in the Theorem assumptions implies

$$
\sup_{z \in \mathcal{C}}\mathbb{E}\left[\langle F(z), \widehat{z}^K - z\rangle\right] = \mathcal{O}\left(\frac{L\Omega_{\mathcal{C}}^2}{K} + \frac{\sigma\Omega_{\mathcal{C}}}{\sqrt{MK}} + \frac{\sqrt{L\Omega_{\mathcal{C}}^3\sqrt{\Delta}}}{K^{3/4}} + \sqrt{\frac{((\Omega + \Omega_{\mathcal{C}})L\sqrt{\Delta} + \Delta)\Omega_{\mathcal{C}}^2}{K}}\right).
$$

$\qquad\square$

### C.4 Proof of Theorem 4.1, Non-Monotone Case

The proof relies on the general per-iterate estimate in Section C.2, which we refine for the purposes of this section. We start with the same estimate (24):

$$
\mathbb{E}\left[\|\bar{z}^{k+1} - z^*\|^2\right] \leq \mathbb{E}\left[\|\bar{z}^k - z^*\|^2\right] - \mathbb{E}\left[\|\bar{z}^{k+1/3} - \bar{z}^k\|^2\right]
$$
$$
- 2\gamma\mathbb{E}\left[\langle \bar{g}^{k+1/3}, \bar{z}^{k+1/3} - z^*\rangle\right] + \gamma^2\mathbb{E}\left[\|\bar{g}^{k+1/3} - \bar{g}^k\|^2\right]. \tag{48}
$$

We use the same Lemma C.4 to bound the last term in the r.h.s., and the following counterpart of Lemma C.3 to deal with the last but one term.

**Lemma C.10.** *Under Assumptions 3.1, 3.2(NM), 3.3 it holds that*

$$
-2\gamma\mathbb{E}\left[\langle \bar{g}^{k+1/3}, \bar{z}^{k+1/3} - z^*\rangle\right] \leq 2\gamma L\sqrt{\mathbb{E}\left[\|\bar{z}^k - z^*\|^2\right]}\sqrt{\mathbb{E}\left[\text{Err}(k+1/3)\right]}
$$
$$
+ \gamma L\mathbb{E}\left[\|\bar{z}^{k+1/3} - \bar{z}^k\|^2\right] + \gamma L\mathbb{E}\left[\text{Err}(k+1/3)\right]. \tag{49}
$$

**Proof:** First of all, we use the independence of all random vectors $\xi^i = (\xi_1^i, \ldots, \xi_m^i)$ and take only the conditional expectation $\mathbb{E}_{\xi^{k+1/3}}$ w.r.t. the vector $\xi^{k+1/3}$, conditioned on all the other randomness. This gives us the following chain of inequalities:

$$
-2\gamma\mathbb{E}\left[\langle \bar{g}^{k+1/3}, \bar{z}^{k+1/3} - z^*\rangle\right] \overset{(17)}{=} -2\gamma\mathbb{E}\left[\left\langle \frac{1}{M}\sum_{m=1}^{M}\mathbb{E}_{\xi^{k+1/3}}[F_m(z_m^{k+1/3}, \xi_m^{k+1/3})], \bar{z}^{k+1/3} - z^*\right\rangle\right]
$$

$$\overset{(4)}{=} -2\gamma\mathbb{E}\left[\left\langle \frac{1}{M}\sum_{m=1}^{M} F_m(z_m^{k+1/3}), \bar{z}^{k+1/3} - z^* \right\rangle\right]$$

$$= -2\gamma\mathbb{E}\left[\left\langle \frac{1}{M}\sum_{m=1}^{M} F_m(\bar{z}^{k+1/3}), \bar{z}^{k+1/3} - z^* \right\rangle\right]$$

$$\quad + 2\gamma\mathbb{E}\left[\left\langle \frac{1}{M}\sum_{m=1}^{M} [F_m(\bar{z}^{k+1/3}) - F_m(z_m^{k+1/3})], \bar{z}^{k+1/3} - z^* \right\rangle\right]$$

$$= -2\gamma\mathbb{E}\left[\left\langle F(\bar{z}^{k+1/3}), \bar{z}^{k+1/3} - z^* \right\rangle\right]$$

$$\quad + 2\gamma\mathbb{E}\left[\left\langle \frac{1}{M}\sum_{m=1}^{M} [F_m(\bar{z}^{k+1/3}) - F_m(z_m^{k+1/3})], \bar{z}^{k+1/3} - z^* \right\rangle\right]$$

$$\overset{(NM)}{\leq} 2\gamma\mathbb{E}\left[\left\langle \frac{1}{M}\sum_{m=1}^{M} [F_m(\bar{z}^{k+1/3}) - F_m(z_m^{k+1/3})], \bar{z}^{k+1/3} - z^* \right\rangle\right]$$

$$\leq 2\gamma\mathbb{E}\left[\|\bar{z}^{k+1/3} - z^*\| \cdot \left\|\frac{1}{M}\sum_{m=1}^{M} F_m(\bar{z}^{k+1/3}) - F_m(z_m^{k+1/3})\right\|\right]$$

$$\leq 2\gamma\mathbb{E}\left[\|\bar{z}^{k+1/3} - z^*\| \cdot \frac{1}{M}\sum_{m=1}^{M} \left\|F_m(\bar{z}^{k+1/3}) - F_m(z_m^{k+1/3})\right\|\right]$$

$$\overset{(L)}{\leq} 2\gamma L\mathbb{E}\left[\|\bar{z}^{k+1/3} - z^*\| \cdot \frac{1}{M}\sum_{m=1}^{M} \left\|z_m^{k+1/3} - \bar{z}^{k+1/3}\right\|\right]$$

$$\leq 2\gamma L\mathbb{E}\left[\|\bar{z}^k - z^*\| \cdot \frac{1}{M}\sum_{m=1}^{M} \left\|z_m^{k+1/3} - \bar{z}^{k+1/3}\right\|\right]$$

$$\quad + 2\gamma L\mathbb{E}\left[\|\bar{z}^{k+1/3} - \bar{z}^k\| \cdot \frac{1}{M}\sum_{m=1}^{M} \left\|z_m^{k+1/3} - \bar{z}^{k+1/3}\right\|\right]$$

$$\overset{(14),(12)}{\leq} 2\gamma L\sqrt{\mathbb{E}\left[\|\bar{z}^k - z^*\|^2\right]} \cdot \sqrt{\mathbb{E}\left[\left(\frac{1}{M}\sum_{m=1}^{M} \left\|z_m^{k+1/3} - \bar{z}^{k+1/3}\right\|\right)^2\right]}$$

$$\quad + \gamma L\mathbb{E}\left[\|\bar{z}^{k+1/3} - \bar{z}^k\|^2\right] + \gamma L\mathbb{E}\left[\left(\frac{1}{M}\sum_{m=1}^{M} \|\bar{z}^{k+1/3} - z_m^{k+1/3}\|\right)^2\right].$$

It is easy to see that, by convexity of the squared norm,

$$\mathbb{E}\left[\left(\frac{1}{M}\sum_{m=1}^{M} \|\bar{z}^{k+1/3} - z_m^{k+1/3}\|\right)^2\right] \leq \mathbb{E}\left[\frac{1}{M}\sum_{m=1}^{M} \|\bar{z}^{k+1/3} - z_m^{k+1/3}\|^2\right] \overset{(23)}{=} \mathbb{E}\mathrm{Err}(k+1/3).$$

This completes the proof.

$$\qquad\qquad\qquad\qquad\qquad\qquad\qquad\qquad\qquad\qquad\qquad\qquad\qquad\qquad\qquad \square$$

We next move to the refinement of the general per-iterate estimate (48). Namely, we substitute (49) and (27) into (48), and obtain the following counterpart of (28):

$$\mathbb{E}\left[\|\bar{z}^{k+1} - z^*\|^2\right] \leq \mathbb{E}\left[\|\bar{z}^k - z^*\|^2\right] - \mathbb{E}\left[\|\bar{z}^{k+1/3} - \bar{z}^k\|^2\right]$$

$$\quad + 2\gamma L\sqrt{\mathbb{E}\left[\|\bar{z}^k - z^*\|^2\right]}\sqrt{\mathbb{E}\left[\mathrm{Err}(k+1/3)\right]}$$

$$\quad + \gamma L\mathbb{E}\left[\|\bar{z}^{k+1/3} - \bar{z}^k\|^2\right] + \gamma L\mathbb{E}\left[\mathrm{Err}(k+1/3)\right]$$

$$+ \gamma^2 \left( 5L^2 \mathbb{E}\left[\|\bar{z}^{k+1/3} - \bar{z}^k\|^2\right] + \frac{10\sigma^2}{M} + 5L^2 \mathbb{E}\left[\text{Err}(k+1/3)\right] + 5L^2 \mathbb{E}\left[\text{Err}(k)\right]\right).$$

Since, by the Theorem assumptions, $\gamma \leq \frac{1}{5L}$, we have

$$\frac{1}{2}\mathbb{E}\left[\|\bar{z}^{k+1/3} - \bar{z}^k\|^2\right] \leq \mathbb{E}\left[\|\bar{z}^k - z^*\|^2\right] - \mathbb{E}\left[\|\bar{z}^{k+1} - z^*\|^2\right]$$

$$+ 2\gamma L\sqrt{\mathbb{E}\left[\|\bar{z}^k - z^*\|^2\right]}\sqrt{\mathbb{E}\left[\text{Err}(k+1/3)\right]}$$

$$+ (5\gamma^2 L^2 + \gamma L)\mathbb{E}\left[\text{Err}(k+1/3)\right] + 5\gamma^2 L^2 \mathbb{E}\left[\text{Err}(k)\right] + \frac{10\gamma^2\sigma^2}{M}. \qquad (50)$$

We next elaborate the term

$\mathbb{E}\left[\|\bar{z}^{k+1/3} - \bar{z}^k\|^2\right]$

$$= \gamma^2 \mathbb{E}\left[\left\|\frac{1}{M}\sum_{m=1}^{M}(F_m(z_m^k, \xi_m^k) - F_m(z_m^k) + F_m(z_m^k) - F_m(\bar{z}^k) + F_m(\bar{z}^k))\right\|^2\right]$$

$$\overset{(13)}{\geq} \frac{\gamma^2}{2}\mathbb{E}\left\|F(\bar{z}^k)\right\|^2 - \gamma^2 \mathbb{E}\left[\left\|\frac{1}{M}\sum_{m=1}^{M}(F_m(z_m^k, \xi_m^k) - F_m(z_m^k) + F_m(z_m^k) - F_m(\bar{z}^k))\right\|^2\right]$$

$$\overset{(13)}{\geq} \frac{\gamma^2}{2}\mathbb{E}\left\|F(\bar{z}^k)\right\|^2 - 2\gamma^2 \mathbb{E}\left[\left\|\frac{1}{M}\sum_{m=1}^{M}F_m(z_m^k, \xi_m^k) - F_m(z_m^k)\right\|^2\right] - 2\gamma^2 \mathbb{E}\left[\left\|\frac{1}{M}\sum_{m=1}^{M}F_m(z_m^k) - F_m(\bar{z}^k)\right\|^2\right]$$

$$\overset{(L)}{\geq} \frac{\gamma^2}{2}\mathbb{E}\left\|F(\bar{z}^k)\right\|^2 - \frac{2\gamma^2\sigma^2}{M} - \frac{2\gamma^2 L^2}{M}\sum_{m=1}^{M}\mathbb{E}\left[\|z_m^k - \bar{z}^k\|^2\right]$$

$$= \frac{\gamma^2}{2}\mathbb{E}\left\|F(\bar{z}^k)\right\|^2 - \frac{2\gamma^2\sigma^2}{M} - 2\gamma^2 L^2 \mathbb{E}\left[\text{Err}(k)\right].$$

Substituting this into (50) gives

$$\frac{\gamma^2}{4}\mathbb{E}\left[\|F(\bar{z}^k)\|^2\right] \leq \mathbb{E}\left[\|\bar{z}^k - z^*\|^2\right] - \mathbb{E}\left[\|\bar{z}^{k+1} - z^*\|^2\right]$$

$$+ 2\gamma L\sqrt{\mathbb{E}\left[\|\bar{z}^k - z^*\|^2\right]}\sqrt{\mathbb{E}\left[\text{Err}(k+1/3)\right]}$$

$$+ (\gamma L + 5\gamma^2 L^2)\mathbb{E}\left[\text{Err}(k+1/3)\right] + 6\gamma^2 L^2 \mathbb{E}\left[\text{Err}(k)\right] + \frac{11\gamma^2\sigma^2}{M}.$$

This is a refined general bound for the per-iteration progress in the non-monotone setting. Applying Lemma C.7 for the consensus error terms, we get

$$\frac{\gamma^2}{4}\mathbb{E}\left[\|F(\bar{z}^k)\|^2\right] \leq \mathbb{E}\left[\|\bar{z}^k - z^*\|^2\right] - \mathbb{E}\left[\|\bar{z}^{k+1} - z^*\|^2\right]$$

$$+ 2\gamma L\sqrt{\mathbb{E}\left[\|\bar{z}^k - z^*\|^2\right]}\sqrt{\frac{8\gamma^2\tau}{p}\cdot\left(\frac{225D^2\tau}{p} + 25\sigma^2\right)}$$

$$+ \gamma^2 \left(\frac{11\sigma^2}{M} + \frac{8(\gamma L + 11\gamma^2 L^2)\tau}{p}\cdot\left(\frac{225D^2\tau}{p} + 25\sigma^2\right)\right).$$

Finally, summation over all $k$ from $0$ to $K$ and averaging gives the following bound.

$$\mathbb{E}\left[\frac{1}{K+1}\sum_{k=0}^{K}\|F(\bar{z}^k)\|^2\right] \leq \frac{4\|z^0 - z^*\|^2}{\gamma^2(K+1)} - \frac{4\mathbb{E}\left[\|z^{K+1} - z^*\|^2\right]}{\gamma^2(K+1)} + \frac{44\sigma^2}{M}$$

$$+ \sqrt{\frac{32L^2\tau}{p}\cdot\left(\frac{225D^2\tau}{p} + 25\sigma^2\right)}\cdot\frac{1}{K+1}\sum_{k=0}^{K}\sqrt{\mathbb{E}\left[\|\bar{z}^k - z^*\|^2\right]}$$

$$+ \frac{8(\gamma L + 11\gamma^2 L^2)\tau}{p}\cdot\left(\frac{225D^2\tau}{p} + 25\sigma^2\right). \qquad (51)$$

Our next goal is to analyze this bound in two situations: bounded iterates and unbounded iterates.

### C.4.1 Unbounded Iterates

First, we consider the general case when the iterates $\bar{z}^k$ are not assumed to be bounded. We carefully analyze this sequence and prove that it can not go too far from the solution to the variational inequality. This allows us to obtain the final convergence rate bound. To that end, first, we write the following corollary of (51):

$$\mathbb{E}\left[\|\bar{z}^{K+1} - z^*\|^2\right] \leq \|z^0 - z^*\|^2 + \frac{11\gamma^2(K+1)\sigma^2}{M}$$
$$+ \sqrt{\frac{2\gamma^4 L^2 \tau}{p} \cdot \left(\frac{225 D^2 \tau}{p} + 25\sigma^2\right)} \cdot \sum_{k=0}^{K} \sqrt{\mathbb{E}\left[\|\bar{z}^k - z^*\|^2\right]}$$
$$+ \frac{\gamma^2(K+1)(\gamma L + 11\gamma^2 L^2)\tau}{2p} \cdot \left(\frac{225 D^2 \tau}{p} + 25\sigma^2\right).$$

Next, we apply Lemma C.9 with $R_k = \sqrt{\mathbb{E}\left[\|\bar{z}^k - z^*\|^2\right]}$, $b = \sqrt{\frac{2\gamma^4 L^2 \tau}{p} \cdot \left(\frac{225 D^2 \tau}{p} + 25\sigma^2\right)}$, $a_k = \frac{\gamma^2(\gamma L + 11\gamma^2 L^2)\tau}{2p} \cdot \left(\frac{225 D^2 \tau}{p} + 25\sigma^2\right) + \frac{11\gamma^2\sigma^2}{M}$, $a_0 = \|z^0 - z^*\|^2 + \frac{\gamma^2(\gamma L + 11\gamma^2 L^2)\tau}{2p} \cdot \left(\frac{225 D^2 \tau}{p} + 25\sigma^2\right) + \frac{11\gamma^2\sigma^2}{M}$, and get

$$\sum_{k=0}^{K} a_k + b \sum_{k=1}^{K} R_k \leq \left(\sqrt{\sum_{k=0}^{K} a_k} + \sqrt{2}b(K+1)\right)^2 \leq 2\sum_{k=0}^{K} a_k + 4b^2(K+1)^2,$$

which gives

$$\sum_{k=1}^{K} R_k \leq \frac{1}{b}\sum_{k=0}^{K} a_k + 4b(K+1)^2.$$

Substituting this in (51) with the same notation, we have

$$\frac{\gamma^2(K+1)}{4}\mathbb{E}\left[\frac{1}{K+1}\sum_{k=0}^{K} \|F(\bar{z}^k)\|^2\right] \leq \sum_{k=0}^{K} a_k + b\left(\frac{1}{b}\sum_{k=0}^{K} a_k + 4b(K+1)^2\right),$$

and, hence,

$$\mathbb{E}\left[\frac{1}{K+1}\sum_{k=0}^{K} \|F(\bar{z}^k)\|^2\right] \leq \frac{8}{\gamma^2(K+1)}\sum_{k=0}^{K} a_k + \frac{16b^2(K+1)}{\gamma^2}.$$

Finally, we get

$$\mathbb{E}\left[\frac{1}{K+1}\sum_{k=0}^{K} \|F(\bar{z}^k)\|^2\right] = \mathcal{O}\left(\frac{\|z^0 - z^*\|^2}{\gamma^2(K+1)} + \frac{(\gamma L + \gamma^2 L^2)\tau}{p} \cdot \left(\frac{D^2\tau}{p} + \sigma^2\right)\right.$$
$$\left. + \frac{\sigma^2}{M} + \frac{(K+1)\gamma^2 L^2 \tau}{p} \cdot \left(\frac{D^2\tau}{p} + \sigma^2\right)\right).$$

As before, we denote $\Delta := 32 \cdot \frac{\tau}{p} \cdot \left(\frac{225 D^2 \tau}{p} + 25\sigma^2\right)$. Our choice $\gamma = \min\left\{\frac{1}{5L}, \left(\frac{\|z^0 - z^*\|^2}{(K+1)^2 L^2 \Delta}\right)^{1/4}\right\}$ in the Theorem assumptions further implies

$$\mathbb{E}\left[\frac{1}{K+1}\sum_{k=0}^{K} \|F(\bar{z}^k)\|^2\right] = \mathcal{O}\left(\frac{L^2\|z^0 - z^*\|^2}{K} + L\|z^0 - z^*\|\sqrt{\Delta}\right.$$
$$\left. + \frac{\sigma^2}{M} + \frac{\sqrt{L}\|z^0 - z^*\|\Delta^{3/4}}{\sqrt{K}}\right).$$

### C.4.2 Bounded iterates

Under the additional assumption that $\|z^*\| \leq \Omega$ and $\|\bar{z}^k\| \leq \Omega$, using (51), we obtain

$$\mathbb{E}\left[\frac{1}{K+1}\sum_{k=0}^{K}\|F(\bar{z}^k)\|^2\right] = \mathcal{O}\left(\frac{\|z^0 - z^*\|^2}{\gamma^2(K+1)} + \frac{(\gamma L + \gamma^2 L^2)\tau}{p} \cdot \left(\frac{D^2\tau}{p} + \sigma^2\right)\right.$$
$$\left. + \frac{\sigma^2}{M} + \sqrt{\frac{L^2\Omega^2\tau}{p} \cdot \left(\frac{D^2\tau}{p} + \sigma^2\right)}\right).$$

Our choice $\gamma = \min\left\{\frac{1}{5L}, \left(\frac{\Omega^2}{(K+1)L\Delta}\right)^{1/3}\right\}$ in the Theorem assumptions further implies

$$\mathbb{E}\left[\frac{1}{K+1}\sum_{k=0}^{K}\|F(\bar{z}^k)\|^2\right] = \mathcal{O}\left(\frac{L^2\Omega^2}{K} + \frac{\sigma^2}{M} + \frac{(L\Omega\Delta)^{2/3}}{K^{1/3}} + L\Omega\sqrt{\Delta}\right).$$

$\square$

## D  Anytime Convergence via a Restart Technique

In this section, we propose a simple procedure that gives our method more flexibility by avoiding the fixed budget $K$ for the number of iterations that needs to be set before the start of the method. We start with a generic interpretation of Theorem 4.1 that combines all the cases considered in the theorem. In all the cases, there is some optimality measure $\rho(K)$, e.g., $\rho(K) = \mathbb{E}\left[\|\bar{z}^{K+1} - z^*\|^2\right]$ in the strongly-monotone case. Further, in all the cases there is some function $\Xi(K)$ which bounds $\rho(K)$ from above after $K$ iterations. Theorem 4.1 states that if we fix the budget of $K$ iterations and set the stepsize $\gamma(K)$, then after $K$ iterations it is guaranteed that $\rho(K) \leq \Xi(K)$. Let us refer to the iterations of Algorithm 1 as basic iterations. We organize the restart procedure as follows. We construct a sequence of the budgets $K_t = 2^t$ for $t \geq 0$. For each restart $t$ we set the stepsize $\gamma(K_t)$, run the algorithm for $K_t$ basic iterations and use the obtained point as a warm-start for the next restart. We can also use the same starting point for all the restarts.

Let us now assume that the algorithm has made $N$ basic iterations. This means that it made at least $T = \lfloor\log_2(N+1)\rfloor - 1$ restarts. Since at the end of the last restart it made $K_T$ basic iterations with the stepsize $\gamma(K_T)$, we obtain, by Theorem 4.1, that we guarantee that

$$\rho(K_T) \leq \Xi(K_T) = \Xi\left(2^T\right) = \Xi\left(2^{\lfloor\log_2(N+1)\rfloor - 1}\right) = \Xi\left(\mathcal{O}(N)\right).$$

Since in all the cases in Theorem 4.1, we have that the dependence of $\Xi(K)$ on $K$ is either exponential or polynomial, we obtain that $\rho(K_T) = \Xi\left(\mathcal{O}(N)\right) = \mathcal{O}\left(\Xi(N)\right)$. Thus, we have obtained an anytime-convergent algorithm with the convergence rates, up to constant factors, similar to that of Algorithm 1. This algorithm does not require to fix the number of basic steps $K$ in advance.