# OpenReview forum: "Decentralized Local Stochastic Extra-Gradient for Variational Inequalities"
_NeurIPS.cc/2022/Conference — NeurIPS 2022 Accept_

### Official Review · Reviewer_nETb · 2022-07-06

**Rating:** 7
**Confidence:** 3
**Soundness:** 3 good
**Presentation:** 3 good
**Contribution:** 3 good

**Summary:**

In this work, the authors address the problem of solving minty variational inequality in decentralized optimization. They extend extra-gradient to this setup and prove convergence results under different assumptions on the operator (strong monotonicity, monotonicity, existence of Minty solution) and a weak assumption on the network topology (expected multi-step contraction). These theoretical results are further complemented by experiments on a syntactic problem and GANs.

**Questions:**

1. In (7) and (8) the supremum is outside the expectation while for stochastic VIs it is more common to have supremum inside the expectation and there is a standard process to achieve this in the centralized case. Actually, it is more meaningful to have the supremum inside the expectation because the gap function should be used as a measure for every realization. Therefore, I am wondering if we can get a result of this kind here or if there is any technical difficulty that makes the authors to present results like this.

2. In the experiments, the authors observe linear convergence in the bilinear case when there is no noise. I understand why this should be the case in the centralized case. However, in the decentralized case with data heterogeneity, the presence of network error should prevent the iterates from converging, as we see in the strongly monotone setup. Could you please explain why this is not observed in the bilinear case?

**Limitations:**

Fine

**Strengths And Weaknesses:**

This work nicely complements the existing literature by providing an exhaustive analysis on decentralized extra-gradient for solving VI problems. On the positive side, the assumption made in this work about the network allows us to take into account various communication schemes. Moreover, their convergence rates feature explicit dependence on different constants of the problem and the network.

On the downside, the proposed algorithm can only partially handle the case of non-monotone operators with minty solutions. In fact, both $\sigma$, the standard deviation of the noise, and $D$, the variability among nodes appear in the bounds (9) and (10) in terms that do not depend on $K$. Even if $D=0$, one still needs to increase the batch size to get better accuracy. This is probably a problem that already needs to be addressed in the centralized case.

I also feel that it is not clear what the authors want to convey through section 5.2. We can probably also get the same types of results with other decentralized algorithms that are mentioned in related work.

---

> ### Author Response · Authors · 2022-07-30
> **Response to Reviewer nETb**
>
> We thank Reviewer **nETb** for the work! We are glad that on the whole the paper received a positive reaction from Reviewer. Next, we try to resolve the issue that Reviewer noted.
>
> > **On the downside, the proposed algorithm can only partially handle the case of non-monotone operators with minty solutions.** In fact, both \sigma, the standard deviation of the noise, and D, the variability among nodes appear in the bounds (9) and (10) in terms that do not depend on K. Even if D=0 , one still needs to increase the batch size to get better accuracy. This is probably a problem that already needs to be addressed in the centralized case.
>
> We agree with Reviewer. Note that the method from the work-competitor [1] has the same disadvantage.
>
> > **I also feel that it is not clear what the authors want to convey through section 5.2.** We can probably also get the same types of results with other decentralized algorithms that are mentioned in related work.
>
> Among the papers for VIs and saddle point problems, there are no works that consider decentralized methods with local updates. In Section 5.2 we need exactly such methods, this is because we use different decentralized networks and communicate only once per epoch (or less frequently). It means that one (or less frequently) time per epoch we have some connection network (full graph or clusters), but in other iterations the connection graph is empty. We check how reducing the edges in the connection graph (full graph vs clusters) or increasing the number of local iterations (full graph at each epoch vs full graph at each 5th epoch) affect the learning process.
>
> > **In (7) and (8) the supremum is outside the expectation** while for stochastic VIs it is more common to have supremum inside the expectation and there is a standard process to achieve this in the centralized case. Actually, it is more meaningful to have the supremum inside the expectation because the gap function should be used as a measure for every realization. Therefore, I am wondering if we can get a result of this kind here or if there is any technical difficulty that makes the authors to present results like this.
>
> 1) We agree that the criterion that Reviewer suggested is better than the one we use. Unfortunately, we could not obtain this result without additional tricks. This is an interesting question for future works.
>
> 2) Note that it is in fact possible to obtain convergence for monotone inequalities on the argument ($|| z^k - z^*||^2$). Such guarantees even better than the discussed supremum. To do this we can use the classical regularization trick. This trick usually makes convex problems strongly convex. The idea is that the regularization parameter is so small ($\sim \varepsilon$, accuracy of the solution) that it almost does not spoil the quality of the solution. In detail, a monotone operator $F$ is replaced with strongly monotone $F + e_k T$, where $T$ is a strongly monotone operator and $e_k > 0$ is a regularization parameter (depends on the iteration number $k$). If we denote $z^k$ as a solution of the regularized VI, then it is possible to prove [1] that $z^k$ converges to $z^*$ for $e_k \to 0$.
>
> > **In the experiments, the authors observe linear convergence in the bilinear case when there is no noise.** I understand why this should be the case in the centralized case. However, in the decentralized case with data heterogeneity, the presence of network error should prevent the iterates from converging, as we see in the strongly monotone setup. Could you please explain why this is not observed in the bilinear case?
>
> 1) If we understand correctly, this question is related to Figure 1, where in the left plot in the case of zero noise, the method stoped converging after some accuracy (this result is expected and understood by Reviewer). In the right plot it seems that the method converged and didn’t stopped. In fact, the stopping accuracy was just cutted on the plot. The stopping accuracy is very much tied to the randomness of problem generation. In the revision version we will insert a more informative plot to avoid misunderstandings.
>
> 2) We also ask Reviewer to pay attention to Figure 6 (Appendix A.2) where we compare convergence for the bilinear problem with constant and decreasing steps. In the case of zero noise, the constant step method reaches a certain accuracy and stops (this is due to heterogeneity), the method with the decreasing step converge slower, but much deeper.
>
> [1] A.B. Bakushinskii and B.T. Polyak. On the solution of variational inequalities

---

> > ### Comment · Reviewer_nETb · 2022-08-03
> > **Thanks for the clairifaction**
> >
> > I would like to thank the authors for the clarification. I have increased my score accordingly.

---

> > > ### Author Response · Authors · 2022-08-04
> > > **Thanks for the review**
> > >
> > > We thank Reviewer again for the review, the time, and the positive feedback on our work!

---

### Official Review · Reviewer_dmBR · 2022-07-10

**Rating:** 3
**Confidence:** 4
**Soundness:** 3 good
**Presentation:** 1 poor
**Contribution:** 2 fair

**Summary:**

This paper studies distributed stochastic variational inequalities (VIs) problems on unbounded domains while the data on different devices might be heterogeneous (non-IID). The authors propose a new algorithm that (1) the gossip matrix is drawn from some distribution in very communication and (2) multiple updates are made on every device between two communications. The authors prove that the proposed method speedups in all strong-monotone, monotone, and non-monotone cases. Comprehensive experiments show that the proposed method have faster speeds in solving distributed stochastic saddle-point problems (SPP), e.g., to training Deep Generative Adversarial Networks (GANs).

**Questions:**

Please clarify the difference and advantages from the following papers:

A. H. Sayed, Diffusion adaptation over networks, vol. 3, pp. 323-453, 2014.

A. H. Sayed, S. Y. Tu, J. Chen, et al, Diffusion strategies for adaption and learning over networks: an examination of distributed strategies and network behavior, IEEE Signal Processing Maganize, vol. 30, no. 3, pp. 155-171, 2013.

S. Y. Xie, L. Guo, Analysis of distributed adaptive filters based on diffusion strategies over sensor networks, IEEE Transactions on Automatic Control, vol. 63, no. 11, pp. 3643-3648, 2018.

S.A. Alghunaim, K. Yuan, A unified and refined convergence analysis for non-convex decentralized learning. IEEE Transactions on Signal Processing. 2022.

A.S. Matveev, M. Almodarresi, R. Ortega, A. Pyrkin, S. Xie. Diffusion-based Distributed Parameter Estimation Through Directed Graphs with Switching Topology: Application of Dynamic Regressor Extension and Mixing. IEEE Transactions on Automatic Control. 2021.

**Limitations:**

The authors did not discuss the limitations and potential negative societal impact of their work, while no significant issues are identified.

**Strengths And Weaknesses:**

Pros:
+ The idea makes sense. Multiple diffusions on devices between two communications can certainly speedups the training.
+ The experiments are nice for a theory-intensive paper. The results are in agreement with the theory.
+ The proofs are comprehensive and clear. I do not find any major flaws.
+ The authors provide the source code of the experiments.

Cons:
- The motivations are quite straightforward. I am concerned it is not novel. Please clarify the difference and advantages from the following papers:

A. H. Sayed, Diffusion adaptation over networks, vol. 3, pp. 323-453, 2014.

A. H. Sayed, S. Y. Tu, J. Chen, et al, Diffusion strategies for adaption and learning over networks: an examination of distributed strategies and network behavior, IEEE Signal Processing Maganize, vol. 30, no. 3, pp. 155-171, 2013.

S. Y. Xie, L. Guo, Analysis of distributed adaptive filters based on diffusion strategies over sensor networks, IEEE Transactions on Automatic Control, vol. 63, no. 11, pp. 3643-3648, 2018.

S.A. Alghunaim, K. Yuan, A unified and refined convergence analysis for non-convex decentralized learning. IEEE Transactions on Signal Processing. 2022.

A.S. Matveev, M. Almodarresi, R. Ortega, A. Pyrkin, S. Xie. Diffusion-based Distributed Parameter Estimation Through Directed Graphs with Switching Topology: Application of Dynamic Regressor Extension and Mixing. IEEE Transactions on Automatic Control. 2021.

- The paper seems written in a rush. Many typos exist.

---

> ### Author Response · Authors · 2022-07-30
> **Response to Reviewer dmBR**
>
> We thank Reviewer **dmBR** for the work! We are glad that Reviewer highlighted 4 pros in our paper. Meanwhile, Reviewer identified 2 cons. Next, we try to solve them.
>
> > **Please clarify the difference and advantages from the following papers**
>
> Hundreds of papers on decentralized algorithms have been published so far. Comparing our results with the results of all these articles is quite problematic.  Therefore, let us limit the scope of the paper to survey only strongly related works.
>
> 1) In our paper we are interested in distributed **variational inequalities** (and **saddle point problems** as a special case) - this is a more general and broad class of problems than minimization problems, moreover, in terms of theory, methods for minimization are not optimal for saddle problems and variational inequalities [6] (Sections 7.2 and 8.2).
>
> 2) **The communication network** over which the devices are distributed can be **time-varying**. Sometimes the network can be empty at all, in particular such a setting can include methods that do **local updates**, without communication.
>
> 3) We give **convergence rates** in three cases: strong-monotone, monotone, non-monotone.
>
> When we wrote the literature survey in our paper we only included methods for distributed variational inequalities and saddle point problems, moreover decentralized methods or methods that support local updates.
>
> All papers that Reviewer gives are not devoted to variational inequalities or saddle point problems. Moreover, some of the papers do not provide the rate of convergence of the methods, but only guarantee that the method converges (but how fast?). In particular,
>
> [1] - only minimization problems are considered (not saddle point problems and variational inequalities), only fixed networks (not time-varying, without local updates), convergence rates are not given
>
> [2] - only minimization problems are considered (not saddle point problems and variational inequalities), convergence rates are not given
>
> [3] - only linear regression are considered (not saddle point problems and variational inequalities), only fixed networks (not time-varying, without local updates), convergence rates are not given
>
> [4] - only minimization problems are considered (not saddle point problems and variational inequalities), only fixed networks (not time-varying, without local updates)
>
> [5] - only linear regression are considered (not saddle point problems and variational inequalities), convergence rates are not given
>
> We do not quite understand why these works should be valuable to us and why we should consider them. These works are more from signal processing than from theoretical optimization.
>
>
> > **The paper seems written in a rush. Many typos exist.**
>
> We kindly disagree with this. We guarantee that the paper was not written in a rush. Moreover, we did proofreading. We would appreciate it if Reviewer would point out typos, it would help make our paper better.
>
>
>
>
> [1] A. H. Sayed, Diffusion adaptation over networks, vol. 3, pp. 323-453, 2014.
>
> [2] A. H. Sayed, S. Y. Tu, J. Chen, et al, Diffusion strategies for adaption and learning over networks: an examination of distributed strategies and network behavior, IEEE Signal Processing Maganize, vol. 30, no. 3, pp. 155-171, 2013.
>
> [3] S. Y. Xie, L. Guo, Analysis of distributed adaptive filters based on diffusion strategies over sensor networks, IEEE Transactions on Automatic Control, vol. 63, no. 11, pp. 3643-3648, 2018.
>
> [4] S.A. Alghunaim, K. Yuan, A unified and refined convergence analysis for non-convex decentralized learning. IEEE Transactions on Signal Processing. 2022.
>
> [5] A.S. Matveev, M. Almodarresi, R. Ortega, A. Pyrkin, S. Xie. Diffusion-based Distributed Parameter Estimation Through Directed Graphs with Switching Topology: Application of Dynamic Regressor Extension and Mixing. IEEE Transactions on Automatic Control. 2021.
>
> [6] I. Goodfellow, NIPS 2016 Tutorial: Generative Adversarial Networks

---

> > ### Comment · Reviewer_dmBR · 2022-08-07
> > **Reply to the authors**
> >
> > Thank you for your response. I agree that this paper has made considerable contributions in the theory now. However, some concerns remain.
> >
> > - The current algorithm includes diffusion strategies on the clients. It has been known diffusion strategies work in distributed learning. Please include a discussion on the differences with the cited papers from the aspects of algorithm design.
> >
> > - I list where the presentation needs to be improved in only the first page as examples:
> >
> >   1. Line 1, Abstract: "(w)e consider distributed stochastic variational inequalities (VIs) on unbounded domains with the problem data being heterogeneous (non-IID) and distributed across many devices." ... with the problem data being ... is a bit weird.
> >
> >   2. Line 9, Abstract: "strongly monotone, monotone, and non-monotone setting" -> "... settings";
> >
> >   3. Line 10, Abstract: "(t)he provided rates have explicit dependence on network characteristics and how it varies with time, data heterogeneity, variance, number of devices, and other standard parameters." What are the "characteristics" here? What does "it" refer to?
> >
> >   4. Line 18, Introduction: "(i)n large scale machine learning (ML) scenarios the training data is often is split over many client devices, such as e.g. geo-distributed datacenters or mobile devices", "large scale -> large-scale";
> >     "is often is split over" -> "is often split over";
> >   "such as e.g." -> "such as";
> >
> >   5. Line 21, Introduction:  "non-centralized" -> "decentralized";
> >
> >   6. Line 31, Introduction:  "or personalization" ->  "and personalization";
> >
> >   7. Line 28 says advances in "development, design, and understand", while line 31 says "all these methods". It is a mismatch;
> >
> >   8. Line 32, Introduction: "single objective loss functions (minimization objective)", "minimization objective" -> "minimization objectives";
> >
> >   9. Line 32, Introduction: "generator and discriminator objective", "objective" -> "objectives";
> >
> >   These are only for page 1. Much more problems exist in the rest of the paper. It definitely needs a thorough revision.

---

> > > ### Author Response · Authors · 2022-08-09
> > > **Response to Reviewer dmBR (Reply to the authors)**
> > >
> > > We are very grateful to Reviewer **dmBR** for the response! We are especially grateful for the careful handling of our text! Using Reviewer's response, we tried to make our paper better.
> > >
> > > > **The current algorithm includes diffusion strategies on the clients. It has been known diffusion strategies work in distributed learning. Please include a discussion on the differences with the cited papers from the aspects of algorithm design.**
> > >
> > > We added this. See the revision of our paper (lines 117 - 124). Unfortunately, due to space limitation, we cannot describe the difference in detail. We give the basic idea. For minimization problems a combination of diffusions and gradient descent is usually used. But since we work with VIs and saddle point problems, we change the gradient descent to the classical method for VIs and saddle point problems - the extragradient method. Accordingly, in contrast to the works mentioned by Reviewer, we consider a combination of the extragradient method and diffusions.
> > >
> > > > **I list where the presentation needs to be improved in only the first page as examples**
> > >
> > > We did another proofreading of the paper and fixed the typos found by Reviewer and some others. We hope that the text of our paper has become better. We have planned a few more proofreadings to finally make sure the quality of the text is good. We thank Reviewer once again for the careful work with the text.

---

### Official Review · Reviewer_n86C · 2022-07-10

**Rating:** 6
**Confidence:** 5
**Soundness:** 2 fair
**Presentation:** 3 good
**Contribution:** 3 good

**Summary:**

This paper proposed a decentralized extra-gradient method with intermittent communications to solve distributed variational inequalities problem, and one important scenario is federated GAN training or federated adversarial training. They consider the setting where clients are connected with a decentralized network and clients only synchronize with its neighborhood. They provide the convergence rate for strongly monotone, monotone and non-monotone cases. The rates make sense. They also provide experiments validating their algorithms' effectiveness and efficiency.

**Questions:**

Please refer to weakness part.

**Limitations:**

This is pure theory paper so I did not see any negative societal impact.

**Strengths And Weaknesses:**

Strengths:
As far as I know, this paper is the first to analyze the extragradient method, a celebrated algorithm in minimax/variational inequality optimization, in decentralized and intermittent communication settings. Due to the rise of federated GAN and adversarial robust training, distributed minimax problems received increasing attention from optimization and ML community. Hence, the theoretical results in this paper have its own value.

Weakness:
My main concern is that the authors assume unbounded parameter domain, but in the monotone setting, we have to assume the parameters are bounded to get a meaningful convergence bound, i.e., in equation (7), they have to assume max_{z,z'}||z-z' || is bounded. I understand that adding projection may collapse existing proof, but assuming bounded domain without projection operator looks unreasonable to me.

---

> ### Author Response · Authors · 2022-07-30
> **Response to Reviewer n86C**
>
> We thank Reviewer **n86C** for the work! We are glad that on the whole the paper received a positive reaction from Reviewer. Next, we try to resolve the issue that Reviewer noted.
>
> > **My main concern is that the authors assume unbounded parameter domain, but in the monotone setting, we have to assume the parameters are bounded to get a meaningful convergence bound**, i.e., in equation (7), they have to assume max_{z,z'}||z-z' || is bounded. I understand that adding projection may collapse existing proof, but assuming bounded domain without projection operator looks unreasonable to me.
>
> It seems that there is a misunderstanding. We do not assume the bounded domain. (7) holds **for any bounded set** $\mathcal{C}$ containing the solution $z^*$. $\mathcal{C}$ is not a feasible set. It is just used for the convergence criterion. This is a standard thing when we work with VIs on unbounded domains, in particular, on $R^d$.  This trick was first used by Y. Nesterov in [1]. We can also give a link to a newer paper that explains this in detail [2] (see 2.5a and Lemma 1). In our work, we also indicate that the set $\mathcal{C}$ is not feasible - see line 268. Thus, there is no contradiction with this assumption and unbounded domain. It is sufficient to assume that the solution is bounded.
>
> [1] Yurii Nesterov. Dual extrapolation and its applications to solving variational inequalities and related problems. Mathematical Programming
>
> [2] Kimon Antonakopoulos, Veronica Belmega, and Panayotis Mertikopoulos. An adaptive mirror- prox method for variational inequalities with singular operators. In Advances in Neural Information Processing Systems 32 (NeurIPS)

---

> > ### Comment · Reviewer_n86C · 2022-08-09
> > **Thanks for authors' response**
> >
> > I would thank authors for their detailed explanation. Now my concern has been resolved, and I increased my score by 1.

---

> > > ### Author Response · Authors · 2022-08-09
> > > **Thank you!**
> > >
> > > We are grateful for raising the score! Thanks for the review and response!

---

### Official Review · Reviewer_zZ6M · 2022-07-11

**Rating:** 5
**Confidence:** 3
**Soundness:** 2 fair
**Presentation:** 2 fair
**Contribution:** 2 fair

**Summary:**

This paper focuses on decentralized local stochastic extra-gradient for variational inequalities.
This paper uses extra-gradient as the main tool to solve saddle point problems which satisfy variational inequalities.
The results of this paper are not surprise but still have their own value.

**Questions:**

No.

**Limitations:**

Yes

**Strengths And Weaknesses:**

The techniques in this paper are regular and the results of this paper are not surprise.
But the results of this paper have their own value.

---

> ### Author Response · Authors · 2022-07-30
> **Response to Reviewer zZ6M**
>
> We thank Reviewer **zZ6M** for the work!
>
> Reviewer was absolutely correct in noting that we present a decentralized modification of extra-gradient with local steps for solving stochastic variational inequalities and saddle point problems. The theoretical analysis is given in the strongly monotone, monotone, and non-monotone cases.
>
> We would be grateful if Reviewer gave us more detailed comments that explains the score and gives us an opportunity to improve the paper.

---

### Author Response · Authors · 2022-08-07
**A kind reminder about rebuttals**

With this message, we would just like to kindly remind Reviewers that we would be happy if Reviewers would participate in the rebuttal discussion process. We are looking forward to hearing from Reviewers **zZ6M**, **n86C** and **dmBR** . We thank Reviewer **nETb** for the responses to the rebuttal.

---

### Meta-Review · Area_Chair_2PEU · 2022-08-30

**Recommendation:** Accept
**Confidence:** Certain

**Metareview:**

The paper studies decentralized local stochastic extra-gradient for variational inequalities.
An extra-gradient method is developed for this problem. Theoretical results are established and complemented by simulations. While there were some concerns about the novelty of the work in the initial review, the authors adequately addressed these comments in their response. While a number of typos were present in the paper, I believe that these can be addressed as a minor revision in the final version. I do encourage the authors to carefully proofread their camera ready submission. The work is of interest to a part of the conference audience and should be accepted.

**Award:**

No

---

### Decision · Program_Chairs · 2022-09-14

Accept